# Energy-Efficient Deep Learning via Update Sampling from a Generalized Gaussian Distribution: An Empirical Study

## Abstract

The computation of loss gradients via backpropagation constitutes a significant portion of the energy consumption in the training of deep learning (DL) models. This paper introduces a simple yet effective method to reduce energy usage during training by leveraging the overparameterization of DL models. Under this assumption, the loss landscape is smooth, and we hypothesize that gradient elements follow a Generalized Gaussian Distribution (GGD). Based on this hypothesis, energy savings are achieved by skipping entire training epochs and estimating gradients by sampling from a GGD. Specifically, parameter updates during skipped epochs are performed by adding GGD-based samples of gradient components to the model parameters from the previous epoch. Furthermore, we present a theorem that provides an upper bound on the expected loss behavior, along with the corresponding convergence rate. We provide extensive empirical validation of our GGD hypothesis across various tasks—image classification, object detection, and image segmentation—using widely adopted DL models. Results show substantial reductions in energy consumption without compromising model performance. Additionally, we evaluate our method on Domain Adaptation (DA), Domain Generalization (DG), and Federated Learning (FL) tasks, observing similar energy savings. To further validate the adaptability of our sampling strategy, we also test it in large language model (LLM) pre-training, demonstrating its effectiveness across diverse settings.

## 1 Introduction

Training deep learning (DL) models is infamous for its energy consumption. This aspect has been discussed and debated due to its impact on global warming and sustainable development. The energy consumption of training and hyperparameter tuning of large DL models was studied in (Strubell et al., 2020). This work recommends AI researchers prioritise the development of energy-efficient hardware and algorithms. A more recent work discusses the energy consumption during inference (Desislavov et al., 2023) and suggests that the energy consumed for inference is growing slower than model training. However, this work also cautions that the energy requirements could escalate if the assumptions made become invalidated by increased penetration of AI-based solutions. Gradient descent-based iterative methods are employed for training DL models. Central to this training process is the forward propagation of data points to compute model predictions, followed by the back-propagation (backprop) (Rumelhart et al., 1986) of the model errors for estimating loss gradients. These gradients are used to update the model parameters. The backprop of model errors is the most energy-consuming operation during training.

In this work, we focus on reducing the number of backprop operations and improving the energy efficiency of DL model training. Also, the problem of improved energy efficiency in the federated learning (FL) framework (Li et al., 2020a) is studied using the same lens. The contributions of this work include:

- a hypothesis that the parameter update distribution is a Generalized Gaussian Distribution (GGD), and its empirical evidence,

- a simple yet effective strategy for saving energy during training based on the above hypothesis, and an extensive empirical validation of the hypothesis using popular DL models on several computer

vision tasks (like Image Classification, Object Detection, Image Segmentation, Domain Adaptation, Domain Generalization) and LLM-Pretraining,

- a stochastic version of federated learning algorithms (McMahan et al., 2017; Li et al., 2020b; Sun et al., 2023) that reduces training rounds, again based on the above hypothesis.

## 2 Related Work

Given the rise of foundation models Liu et al. (2023) with massive energy requirements for training, the need for energy efficiency in model training cannot be overemphasized. The design of energy-efficient methods for training DL models has received significant attention from the hardware perspective. An excellent albeit slightly dated survey of efficient processing methods for DL models is presented in Sze et al. (2017). Esser et al. (2015) proposed an approach for deploying the backprop algorithm in neuromorphic computing design. This is achieved by treating the spikes and discrete synapses as continuous probability distributions and thereby satisfying the requirement of the backprop algorithm. This approach led to significant energy savings on the TrueNorth neuromorphic architecture Merolla et al. (2014). Weight pruning is a popular algorithmic approach to efficient model training without compromising performance. Hoefler et al. (2021) present a comprehensive survey of the various techniques proposed in the literature for sparsifying DL models and Stochastic Gradient Langevin Dynamics (SGLD) Welling & Teh (2011) adds Gaussian noise sampled independently at each iteration, with a fixed zero mean and variance proportional to the step size. This noise is then directly added to the current gradient updates to ensure stochasticity and enable posterior sampling. Recent works Kuo & Madni (2022); Lin et al. (2022) propose green learning by avoiding non-linearities and the backprop algorithm altogether. Gaussian noise injection (GNI) Camuto et al. (2020) act as an explicit regulariser that suppresses high-frequency components in neural network activations, promoting smoother functions and improved generalization. MeZO Malladi et al. (2023) proposes the addition of small Gaussian noise to perturb the parameter updates. It requires only two forward passes through the model to compute the gradient estimate.

Our approach differs from existing methods by retaining backpropagation while adopting a stochastic approach to estimate gradient values, reducing the number of forward and backpropagation operations for greater energy efficiency. In contrast to SGLD Welling & Teh (2011), our method dynamically adapts injected noise based on differences in the model's parameters over time, ensuring better alignment with the parameter space and capturing training variability for improved posterior approximations and generalization. Additionally, unlike MeZO Malladi et al. (2023), our approach is versatile and can be applied to train deep learning models from scratch.

In the FL framework, a central server coordinates the training of clients by aggregating client model parameters (Li et al., 2020a). Each client updates its model parameters locally by training on its private dataset. The clients then share their model parameters with the central server. The server collates the client parameters and computes a set of global weights. These weights are then sent to the clients for use in the next training round. The two-way exchange of model parameters happens over several rounds until the models converge. Let $\boldsymbol{\theta}^{(r)}$ denote the collated parameter vector at the server, and $\boldsymbol{\theta}^{(r,k)}$ represent the client $k$'s parameter vector after the client update at round $r$ (using $\boldsymbol{\theta}^{(r,k)}$ as the initialization). In the FL algorithm, the server update is given by:

$$\boldsymbol{\theta}^{(r+1)} = \frac{\mathcal{D}_i}{|\mathcal{D}|} \sum_{k=1}^{K} \boldsymbol{\theta}^{(r,k)} \tag{1}$$

where $K$ is the number of clients used, $|\mathcal{D}_k|$ is the size of data in client $k$ and $|\mathcal{D}|$ is the size of overall dataset ($|D| = \sum_{k=1}^{K} |\mathcal{D}_k|$). We demonstrate the utility of our hypothesis in the FL framework by reducing the number of communication rounds used to train the clients.

## 3 Proposed Approach

### 3.1 Classification Setting

We operate in the classification setting with the dataset denoted by $\mathcal{D} = \{(\mathbf{x}_1, \mathbf{y}_1), (\mathbf{x}_2, \mathbf{y}_2), \ldots, (\mathbf{x}_N, \mathbf{y}_N)\}$ composed of $N$ data points $\mathbf{x}_i \in \mathcal{X}$ and corresponding labels $\mathbf{y}_i \in \mathcal{Y}$. The data point-label pairs are assumed to be i.i.d. samples drawn from a fixed but unknown distribution $p(\mathbf{x}, \mathbf{y})$. The DL model is denoted by $f(\mathbf{x}; \boldsymbol{\theta}) : \mathcal{X} \to \mathcal{Y}$ where $\mathbf{x}$ is the input data point and the vector $\boldsymbol{\theta}$ represents the model's parameters. The loss function is defined as

$$\mathcal{L}(\boldsymbol{\theta}) = \frac{1}{n} \sum_{i=1}^{n} d(\mathbf{y}_i, f(\mathbf{x}; \boldsymbol{\theta})), \tag{2}$$

where $d(\cdot, \cdot)$ is an appropriately chosen distance function, and $n$ is the number of training samples.

Further, assuming the standard gradient-based iterative model training approach, the parameter update expression is

$$\boldsymbol{\theta}^{(k)} = \boldsymbol{\theta}^{(k-1)} + \boldsymbol{\delta}^{(k)}, \tag{3}$$

where $\boldsymbol{\delta}^{(k)}$ represents the parameter update at iteration $k$.

### 3.2 Hypothesis

In this setting, we empirically hypothesize that for each layer $\ell$ with $n_\ell$ elements, $\boldsymbol{\theta}_{\ell,i}^{(k)}$ denotes the $i^{\text{th}}$ element of the parameter vector $\boldsymbol{\theta}_l^{(k)}$ at layer $\ell$,

$$p(\boldsymbol{\theta}_{\ell,i}^{(k)} | \boldsymbol{\theta}_{\ell,i}^{(k-1)}) \sim \text{GGD}(\mu_\ell^{(k)}, \sigma_\ell^{(k)}, \beta_\ell^{(k)}). \tag{4}$$

In other words, from (3), at each layer $\ell$, the $i^{\text{th}}$ element can be updated as:

$$\boldsymbol{\delta}_{\ell,i}^{(k)} = \left( \boldsymbol{\theta}_{\ell,i}^{(k)} - \boldsymbol{\theta}_{\ell,i}^{(k-1)} \right) \sim \text{GGD}(\mu_\ell^{(k)}, \sigma_\ell^{(k)}, \beta_\ell^{(k)}), \tag{5}$$

where GGD denotes the Generalized Gaussian Distribution, parameterized by $\mu_\ell^{(k)}$, $\sigma_\ell^{(k)}$, and $\beta_\ell^{(k)}$, corresponding to the mean, scale, and shape parameters, respectively, estimated via Maximum Likelihood over $\left( \boldsymbol{\theta}_\ell^{(k)} - \boldsymbol{\theta}_\ell^{(k-1)} \right)$ for each layer $\ell$. Specifically, the GGD is given by

$$f\left( \boldsymbol{\delta}_{\ell,i}^{(k)}; \mu_\ell^{(k)}, \sigma_\ell^{(k)}, \beta_\ell^{(k)} \right) = \frac{\beta_\ell^{(k)}}{2\sigma_\ell^{(k)} \Gamma\left( \frac{1}{\beta_\ell^{(k)}} \right)} \exp\left( -\left| \frac{\boldsymbol{\delta}_{\ell,i}^{(k)} - \mu_\ell^{(k)}}{\sigma_\ell^{(k)}} \right|^{\beta_\ell^{(k)}} \right), \quad \forall i = 1, \cdots, n_\ell, \tag{6}$$

where $\Gamma(\cdot)$ is the gamma function. We provide two supporting arguments for our hypothesis.

- From (3), we obtain:

$$\boldsymbol{\delta}_\ell^{(k)} = \boldsymbol{\theta}_\ell^{(k)} - \boldsymbol{\theta}_\ell^{(k-1)}. \tag{7}$$

  For a mini-batch $B$, this becomes:

$$\boldsymbol{\delta}_\ell^{(k)} |_B = \left( \boldsymbol{\theta}_\ell^{(k)} - \boldsymbol{\theta}_\ell^{(k-1)} \right) |_B \tag{8}$$

  where $\boldsymbol{\delta}_\ell^{(k)} |_B$ denotes the function applied to the gradient computed on mini-batch $B$, with respect to the model parameters $\boldsymbol{\theta}^{(k-1)}$ at layer $\ell$.

  Let $\boldsymbol{\xi}_\ell^B = \boldsymbol{\delta}_\ell^{(k)} |_B - \boldsymbol{\delta}_\ell^{(k)}$ represents the stochastic noise which is the error between the mini-batch gradient and the deterministic gradient. The noise $\boldsymbol{\xi}_\ell^B$ prevents the gradient updates from becoming

zero, ensuring continued exploration of the loss landscape. As optimization progresses, $\boldsymbol{\delta}_\ell^{(k)} \to 0$ near local minima (due to the $L'$-Lipschitz continuity of $\mathcal{L}$), causing $\boldsymbol{\xi}_\ell^B$ to diminish. Consequently, the updates become smaller and unimodal, centered around zero.

We rewrite $\boldsymbol{\delta}_\ell^{(k)}|_B$ in (8) as:

$$\boldsymbol{\delta}_\ell^{(k)}|_B = \boldsymbol{\delta}_\ell^{(k)} + \boldsymbol{\xi}_\ell^B. \tag{9}$$

- Empirical studies by (Şimşekli et al., 2019) show that $\boldsymbol{\xi}_\ell^B$ often exhibits heavy-tailed behavior. According to the Generalized Central Limit Theorem (GCLT) (Gnedenko & Kolmogorov, 1954), the updates $\boldsymbol{\theta}_\ell^{(k)} - \boldsymbol{\theta}_\ell^{(k-1)}$ converge to a stable distribution with heavy tail.

The parameters of the GGD can be estimated using the standard maximum likelihood approach using error vector samples computed over the training epochs. In practice, however, the number of parameters in DL models is prohibitively large for efficient parameter estimation. We propose a simple workaround to overcome this practical challenge. The error vector $\boldsymbol{\delta}^{(k)}$ at the $k^{\text{th}}$ epoch is partitioned into layerwise i.e. $\boldsymbol{\delta}^{(k)} = \left[\boldsymbol{\delta}_1^{(k)}, \boldsymbol{\delta}_2^{(k)}, \cdots, \boldsymbol{\delta}_\ell^{(k)}, \cdots, \boldsymbol{\delta}_L^{(k)}\right]$ (assuming $L$ layers), and modelled at the layer level. Here, $\boldsymbol{\delta}_\ell^{(k)}$ refers to the error sub-vector corresponding the $\ell^{\text{th}}$ layer. A further simplification is to treat all the elements of the $\ell^{\text{th}}$ layer error vector $\boldsymbol{\delta}_\ell^{(k)}$ as i.i.d. random variables whose distribution is given by $\text{GGD}(\mu_\ell^{(k)}, \sigma_\ell^{(k)}, \beta_\ell^{(k)})$. This can be justified by the fact that the inputs and outputs at a given layer have similar dynamic ranges due to operations such as batch normalization and layer normalization. While this may appear to be an over-simplification, it works well in practice.

### 3.3 Energy-Efficient Model Training

We now apply our hypothesis to achieve energy-efficient DL model training. A straightforward way is to estimate the model parameters of the layer-level GGD based on the parameter update $\boldsymbol{\delta}_\ell^{(k)}$ after some $k$ epochs. Given that we ascribe the same GGD to all elements of $\boldsymbol{\delta}_\ell^{(k)}$, its parameters are estimated by the elements of $\boldsymbol{\delta}_\ell^{(k)}$.

During the first iteration of the $(k+1)^{\text{st}}$ epoch, the entire forward and backward propagation are skipped. Instead, the model update happens by sampling the parameter update from the GGD and adding these samples to the parameter $\boldsymbol{\theta}^{(k)}$. This can be expressed as:

$$\boldsymbol{\theta}_\ell^{(k+1)} = \boldsymbol{\theta}_\ell^{(k)} + \hat{\boldsymbol{\delta}}_\ell^{(k)}, \tag{10}$$

where $\hat{\boldsymbol{\delta}}_{\ell,i}^{(k)} \sim \text{GGD}(\boldsymbol{\delta}_\ell^{(k)}; \mu_\ell^{(k)}, \sigma_\ell^{(k)}, \beta_\ell^{(k)})$ samples drawn $n_\ell$ (size of $\boldsymbol{\delta}_\ell^{(k)}$) times from the GGD and reshaped to align with the dimensions of $\boldsymbol{\delta}_\ell^{(k)}$, ensuring uniform size and structure. The update in (10) is carried out for all $L$ layers of the model. The proposed model training approach is summarized in Algorithm 1 we call it **GradSamp**.

A natural question is about the frequency of using (10) for model update. Several strategies could be employed to answer this question. A simple strategy is to make this update periodic and experiment with the period. Another approach is to randomly choose an epoch and apply (10). Yet another strategy is to do sampling based upon half of the delayed epochs and apply (10). All these strategies are explored in section 4.1. We present a theorem for our sampling approach with an upper bound on the expected loss behavior $\Delta_k = \mathbb{E}[\mathcal{L}(\boldsymbol{\theta}^{(k)}) - \mathcal{L}(\boldsymbol{\theta}^{(*)})]$ along with convergence rate as follows:

**Theorem 3.1** (Convergence under GGD noise)**.** *Let $\rho$ be the sampling probability, and suppose that the update error $\boldsymbol{\theta}^{(k)}$ follows a Generalized Gaussian Distribution (GGD), with $\mathbb{E}[\boldsymbol{\theta}^{(k)}] = \mathbf{0}$ and $\mathbb{E}[\|\boldsymbol{\theta}^{(k)}\|] \leq \sigma^{(k)^2}$.*

*Assume the loss function $\mathcal{L}(\boldsymbol{\theta})$ is $\mu'$-strongly convex and $L'$-Lipschitz continuous, with optimal parameter $\boldsymbol{\theta}^{(*)}$. If the learning rate $\eta$ satisfies $0 < \eta < \min\left\{\frac{2}{L'}, \frac{1}{2\mu'(1-\rho)}\right\}$, then the expected loss difference $\Delta_k = \mathbb{E}[\mathcal{L}(\boldsymbol{\theta}^{(k)}) - \mathcal{L}(\boldsymbol{\theta}^{(*)})]$ satisfies:*

$$\Delta_{k+1} \leq \gamma \Delta_k + \frac{\rho L' \sigma^{(k)^2}}{2},$$

*where $\gamma = 1 - (1 - \rho)(2\mu'\eta - L'\mu'\eta^2)$. Consequently, the convergence rate is $\mathcal{O}(\gamma^k)$.*

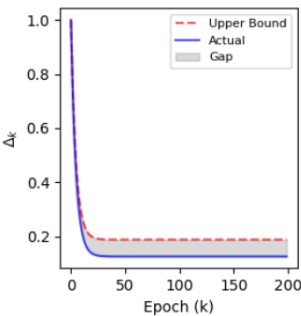

Figure 1: Convergence of $\Delta_K$: Theoretical upper bound vs. actual values.

The detailed proof of Theorem 3.1 is provided in Section A. The plot 1 illustrates the convergence of $\Delta_k$ under the given recursive bound, where the actual values (blue line) exhibit an initial rapid decay, driven by the geometric contraction term $1 - (1 - \rho)(2\mu'\eta - L'\mu'\eta^2)\Delta_k$, followed by a plateau due to the additive noise-dependent term $\frac{\rho L' \sigma^{(k)^2}}{2}$. The theoretical upper bound (red dashed line) consistently stays above the actual values, with the shaded region representing the gap, indicating that while the bound provides a good approximation of the convergence trend,

---

**Algorithm 1 GradSamp** Algorithm

---

**Input:** Model parameters $\boldsymbol{\theta}$, training dataset $\mathcal{D} = \{(\mathbf{x}_i, \mathbf{y}_i)\}_{i=1}^n$, number of layers $L$, parameters per layer $\{n_\ell\}_{\ell=1}^L$, and tolerance $\varepsilon = 0.001$

**Output:** Trained parameters $\boldsymbol{\theta}^{(*)}$

**Initialize:** Epoch counter $k \leftarrow 0$

**while** *stopping condition not met* **do**

    **if** *sampling condition met* **then**

        Compute parameter update error:

        $\boldsymbol{\delta}^{(k-1)} = \boldsymbol{\theta}^{(k-1)} - \boldsymbol{\theta}^{(k-2)}$ ;                        `// From buffer`

        Partition $\boldsymbol{\delta}^{(k-1)}$ by layer:

        $\boldsymbol{\delta}^{(k-1)} = \left[\boldsymbol{\delta}_1^{(k-1)}, \boldsymbol{\delta}_2^{(k-1)}, \ldots, \boldsymbol{\delta}_L^{(k-1)}\right]$

        **for** $\ell \leftarrow 1$ *to* $L$ **do**

            Fit GGD to $\boldsymbol{\delta}_\ell^{(k-1)}$:

            Obtain $\mu_\ell^{(k-1)}$, $\sigma_\ell^{(k-1)}$, and $\beta_\ell^{(k-1)}$ ;        `// By fitting GGD to` $\boldsymbol{\delta}^{(k-1)}$

            Sample $\hat{\boldsymbol{\delta}}_{\ell,i}^{(k-1)} \sim \text{GGD}(\mu_\ell^{(k-1)}, \sigma_\ell^{(k-1)} + \varepsilon, \beta_\ell^{(k-1)})$:   $\forall i = 1, \cdots, n_\ell$

            Generate $n_\ell$ samples and reshape to match $\boldsymbol{\delta}_\ell^{(k-1)}$

            Update parameters for layer $\ell$:

            $\boldsymbol{\theta}_\ell^{(k)} = \boldsymbol{\theta}_\ell^{(k-1)} + \hat{\boldsymbol{\delta}}_\ell^{(k-1)}$

        **end**

    **end**

    **else**

        Save $\boldsymbol{\theta}^{(k-1)}$ in buffer

        Update parameters via backpropagation:

        $\boldsymbol{\theta}^{(k)} = \boldsymbol{\theta}^{(k-1)} + \boldsymbol{\delta}^{(k)}$

    **end**

    $k \leftarrow k + 1$ ;                                     `// Increment epoch counter`

**end**

---

Table 1: Comparison with noise-based methods.

| Paper | Equation | Adaptive Noise | Skipping Iteration |
|---|---|---|---|
| SGLD (Welling & Teh, 2011) | $\boldsymbol{\theta}^{(k+1)} = \boldsymbol{\theta}^{(k)} + \frac{\epsilon_k}{2}\left(\nabla \log p(\boldsymbol{\theta}^{(k)}) + \frac{N}{n}\sum_{i=1}^{n}\nabla \log p(\mathbf{x}_i \mid \boldsymbol{\theta}^{(k)})\right)$ $+ \eta_k;\ \eta_k \sim \mathcal{N}(0, \epsilon_k I)$ | ✗ | ✗ |
| GNI (Camuto et al., 2020) | $\tilde{h}_k(x) = \hat{h}_k(x) + \epsilon_k;\ \epsilon_k \sim \mathcal{N}(0, \sigma_k^2 I)$ | ✗ | ✗ |
| MeZO (Malladi et al., 2023) | $\boldsymbol{\theta}^{(k+1)} = \boldsymbol{\theta}^{(k)} + \frac{\mathcal{L}(\boldsymbol{\theta}^{(k)}+\epsilon z;\mathcal{B})-\mathcal{L}(\boldsymbol{\theta}^{(k)}-\epsilon z;\mathcal{B})}{2\epsilon}z;\quad z \sim \mathcal{N}(0, I)$ | ✗ | ✗ |
| Ours | $\boldsymbol{\theta}_\ell^{(k+1)} = \boldsymbol{\theta}_\ell^{(k)} + \hat{\boldsymbol{\delta}}_\ell^{(k)};\quad \hat{\boldsymbol{\delta}}_{\ell,i}^{(k)} \sim \mathrm{GGD}((\boldsymbol{\theta}_{\ell,i}^{(k-1)} - \boldsymbol{\theta}_{\ell,i}^{(k)});\mu_\ell^{(k)}, \sigma_\ell^{(k)}, \beta_\ell^{(k)})$ | ✓ | ✓ |

## 3.4 Comparison with other noise-based strategies

As shown in Table 1, unlike SGLD (Welling & Teh, 2011), our method does not inject fixed Gaussian noise at each iteration for posterior sampling $p(\mathbf{x} \mid \boldsymbol{\theta}^{(k)})$, nor does it resemble Gaussian noise injection (GNI) (Camuto et al., 2020), which perturbs the activation space $\hat{\boldsymbol{h}}(\mathbf{x})$ to regularize the model. Unlike MeZO (Malladi et al., 2023), which estimates gradients using symmetric perturbations via two forward passes (over batch size $B$ with some perturbation scale $\epsilon$), our approach adaptively estimates parameter updates using temporal differences in the parameter trajectory. Furthermore, existing methods primarily focus on improving model performance without addressing the cost of backpropagation. In contrast, our method explicitly aims to skip backpropagation steps (or iterations), achieving energy efficiency without compromising model accuracy.

## 3.5 Stochastic Federated Learning Algorithms

We found the FL framework to be a natural fit to test our hypothesis in a distributed learning setting. Specifically, we claim that the *elements* of $\boldsymbol{\delta}^{(r)} = \boldsymbol{\theta}^{(r)} - \boldsymbol{\theta}^{(r-1)}$ follow a unimodal GGD (using the notation from (1) and $r$ being the round index). In other words, for the $i^{\text{th}}$ element of the parameter vector $\boldsymbol{\theta}_l^{(r)}$ at layer $\ell$,

$$p(\boldsymbol{\theta}_{\ell,i}^{(r)}|\boldsymbol{\theta}_{\ell,i}^{(r-1)}) \sim \mathrm{GGD}(\mu_\ell^{(r)}, \sigma_\ell^{(r)}, \beta_\ell^{(r)}) \tag{11}$$

We modify the `FedAvg` (McMahan et al., 2017), `FedProx` (Li et al., 2020b) and `FedSpeed` (Sun et al., 2023) algorithms to their stochastic variants where the update rule becomes:

$$\boldsymbol{\theta}_\ell^{(r+1)} = \boldsymbol{\theta}_\ell^{(r)} + \hat{\boldsymbol{\delta}}_\ell^{(r)}, \tag{12}$$

where $\hat{\boldsymbol{\delta}}_{\ell,i}^{(r)} \sim \mathrm{GGD}(\boldsymbol{\delta}_{\ell,i}^{(r)};\mu_\ell^{(r)}, \sigma_\ell^{(r)}, \beta_\ell^{(r)})$ samples are drawn $n_\ell$ (size of $\boldsymbol{\delta}_\ell^{(r)}$) times from the GGD and reshaped to align with the dimensions of $\boldsymbol{\delta}_\ell^{(r)}$, ensuring uniform size and structure.

# 4 Experiments and Results

The proposed hypothesis and energy-based model are empirically validated across a wide range of computer vision tasks, including image classification, object detection, image segmentation, Domain Adaptation (DA), Domain Generalization (DG), and Federated Learning (FL) under both IID and Non-IID settings. Additionally, the method is evaluated on Large Language Model (LLM) training to demonstrate its broader applicability. More technical and implementation details were found in the sections B and C of the supplementary material.

## 4.1 Sampling Strategies

We explored four strategies to trigger the energy-saving mechanism. For periodic sampling (Pe), we used intervals of 5 and 10 epochs. Probabilistic (Pr) sampling drew from a `Bernoulli(`$\rho$`)` distribution with $\rho = 0.2, 0.5$, and $0.7$, sampling gradients whenever a 1 was drawn. In Delayed Period (DP) sampling, no sampling occurred during the first half of the epochs, followed by periodic sampling with intervals of 5 and 10. Similarly, Delayed Random (DR) sampling skipped the first half of the epochs, then applied random sampling with probabilities $\rho = 0.2, 0.5$, and $0.7$.

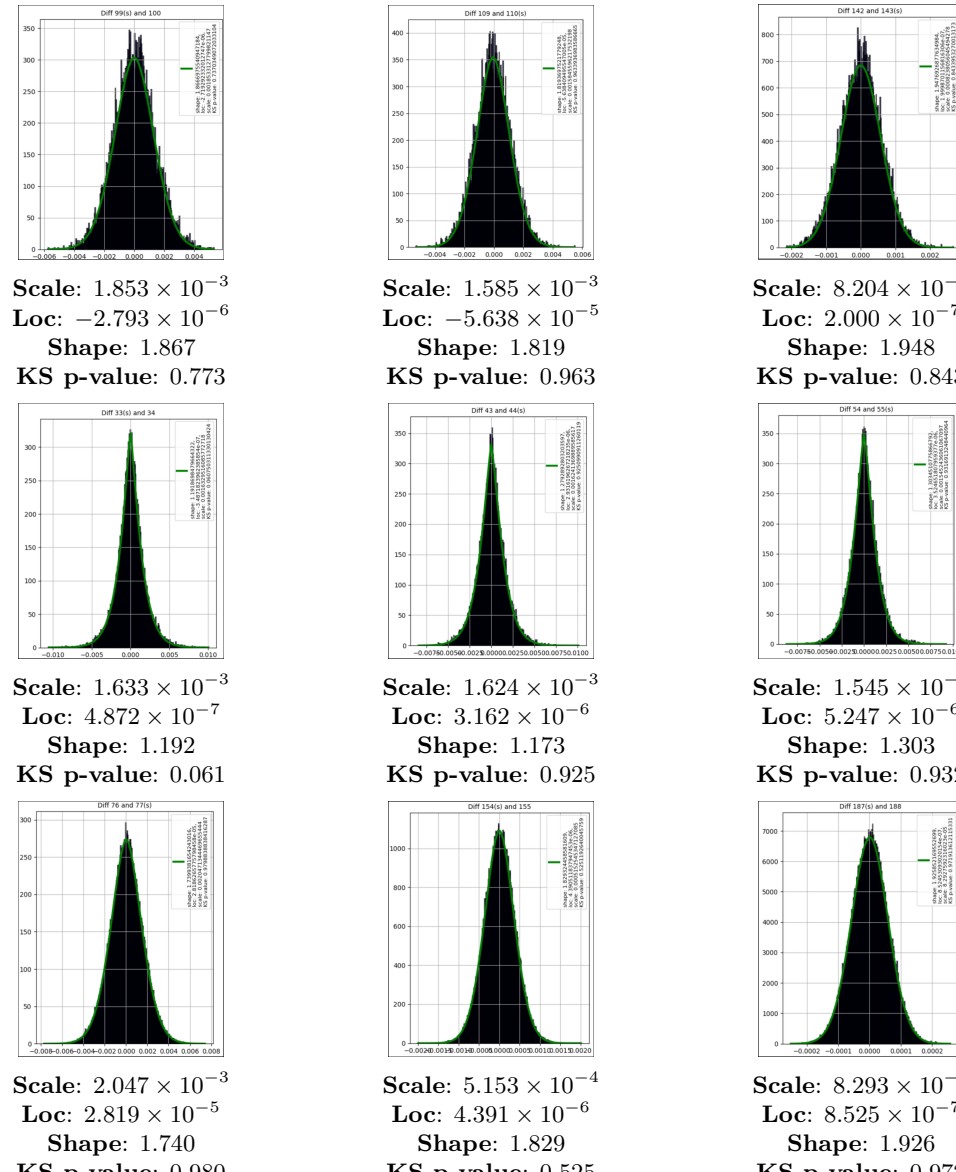

**Scale**: $1.853 \times 10^{-3}$
**Loc**: $-2.793 \times 10^{-6}$
**Shape**: 1.867
**KS p-value**: 0.773

**Scale**: $1.585 \times 10^{-3}$
**Loc**: $-5.638 \times 10^{-5}$
**Shape**: 1.819
**KS p-value**: 0.963

**Scale**: $8.204 \times 10^{-4}$
**Loc**: $2.000 \times 10^{-7}$
**Shape**: 1.948
**KS p-value**: 0.843

**Scale**: $1.633 \times 10^{-3}$
**Loc**: $4.872 \times 10^{-7}$
**Shape**: 1.192
**KS p-value**: 0.061

**Scale**: $1.624 \times 10^{-3}$
**Loc**: $3.162 \times 10^{-6}$
**Shape**: 1.173
**KS p-value**: 0.925

**Scale**: $1.545 \times 10^{-3}$
**Loc**: $5.247 \times 10^{-6}$
**Shape**: 1.303
**KS p-value**: 0.932

**Scale**: $2.047 \times 10^{-3}$
**Loc**: $2.819 \times 10^{-5}$
**Shape**: 1.740
**KS p-value**: 0.980

**Scale**: $5.153 \times 10^{-4}$
**Loc**: $4.391 \times 10^{-6}$
**Shape**: 1.829
**KS p-value**: 0.525

**Scale**: $8.293 \times 10^{-5}$
**Loc**: $8.525 \times 10^{-7}$
**Shape**: 1.926
**KS p-value**: 0.972

Figure 2: Parameter update histograms (fitted with GGD) across three architectures on the Tiny-ImageNet dataset, sampled every 10 epochs. **Top row:** ResNet-50 at the 1st convolution, 2nd stage, and 2nd residual block (Epochs 100, 110, 143). **Middle row:** Swin-T QKV weights at the 1st layer and 2nd block (Epochs 33, 44, 55). **Bottom row:** MLP-Mixer at the 3rd layer, 2nd stage, and 1st fully connected layer (Epochs 77, 155, 188).

## 4.2 Results

**Image Classification:** For the image classification task, we experimented with various baseline networks (He et al., 2016; Liu et al., 2021; Tolstikhin et al., 2021). Qualitative results, shown in the sub-plots present in 2, provide empirical support for our hypothesis. The parameter update histogram at different instances confirm that our hypothesis holds across CNN, Transformer, and Non-CNN/Non-Transformer networks. The standard Kolmogorov-Smirnov (KS)-test (Massey, 1951) goodness of fit with a probability value $p = 5\%$. Upon clear examination of these plots, the mean (loc) and standard deviation (scale) values are approximately zero, and the values of shape parameters are less than 2. This clearly indicates that the parameter updates

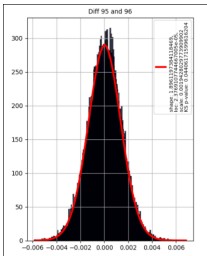 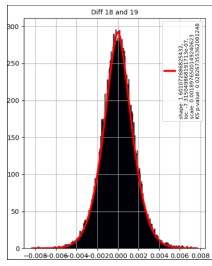 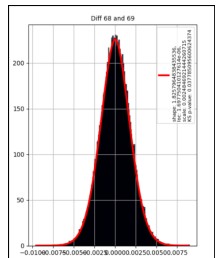

**Scale**: $1.943 \times 10^{-3}$   **Scale**: $1.898 \times 10^{-3}$   **Scale**: $2.485 \times 10^{-3}$
**Loc**: $2.377 \times 10^{-5}$   **Loc**: $-7.315 \times 10^{-7}$   **Loc**: $1.698 \times 10^{-6}$
**Shape**: 1.896   **Shape**: 1.601   **Shape**: 1.826
**KS p-value**: 0.044   **KS p-value**: 0.028   **KS p-value**: 0.038

Figure 3: Failure cases in parameter update histograms (fitted with GGD) across different architectures on the Tiny-ImageNet dataset, sampled every 10 epochs. **Left:** ResNet-50 (2nd Conv, 2nd Stage, 2nd Residual Block, epoch 96). **Middle:** Swin-T (2nd Conv, 2nd Stage, 2nd Layer, epoch 19). **Right:** MLP-Mixer (1st FC Layer, 2nd Stage, 3rd Layer, epoch 96).

are heavy-tailed and fit a nice GGD envelope (in green color). However, even in the failure cases present in figure 3, the parameter update histogram followed a unimodal behavior centered at zero with nearly zero standard deviation, and were fitted by a skewed GGD envelope (in red color). Additionally, it is worth noting that the hypothesis holds better as the number of training epochs increases. Furthermore, the qualitative results reveal that the GGD hypothesis is valid in the convolution layers of CNNs, QKV parameters in transformer models and FC layers in mixer models. This observation aligns with the larger number of parameters present in these layers compared to others.

By examining Table 2, we can estimate energy savings based on the skipped backpropagation operations, as backpropagation accounts for a significant portion of training energy consumption. The proposed approaches are compared against the baseline performance, which is presented in the first row for each model. A key observation is that simple periodic sampling with periods of 5 and 10 epochs leads to energy savings without a significant performance drop across all models, including standard deep learning models, transformer-based architectures, and mixer models. Similarly, the random sampling strategy, where backpropagation is replaced with Generalized Gaussian Distribution (GGD) samples 20% of the time, shows no notable performance degradation. In other words, the proposed method achieves nearly identical model performance while reducing energy consumption by up to 20% compared to the baseline. Given the large scale at which deep learning models are trained, this represents a substantial energy-saving opportunity and highlights the effectiveness of a simple periodic sampling approach. However, a noticeable performance drop is observed when sampling frequencies increase. Importantly, this degradation remains consistent as energy savings increase from 20% to 70%. A similar trend is observed in delayed sampling approaches such as DP and DR, which result in approximately 5% and 20% energy savings, respectively.

**Impact on ImageNet (Deng et al., 2009):** The results in Table 3 demonstrate that incorporating our sampling strategies can significantly reduce computational cost on the ImageNet dataset without compromising model performance. In particular, probabilistic sampling with a probability of 0.2 consistently outperforms the baseline in top-5 and top-10 accuracy across all architectures—ResNet-50, Swin-T, and MLP-Mixer—while achieving up to 20% computational savings. Similarly, periodic sampling with an interval of 10 yields competitive results with minimal accuracy drop and only 10% resource usage reduction. This highlights the effectiveness of using parameter update histogram-guided sampling over GGD-based energy distributions to selectively trigger updates, allowing for better energy-efficiency and scalability in large-scale vision tasks like ImageNet classification.

**Object Detection:** In the object detection setting, we evaluated our sampling strategies using real-time object detectors such as YOLOv7 (Wang et al., 2023) and RT-DETRv2 (Lv et al., 2024), as shown in table 4.

Table 2: Performance comparison of standard models under diverse sampling strategies, including periodic (Pe), probabilistic (Pr), delayed periodic (DP), and delayed random (DR) samplings, along with total FLOPS (in Tera FLOPS (TFLOPS)) over 200 epochs for 3 independent trials. Best values are **bolded**.

| Model | Strategy | CIFAR-10 | | Total TFLOPS | CIFAR-100 | | Total TFLOPS | TINY | | Total TFLOPS |
|---|---|---|---|---|---|---|---|---|---|---|
| | | *Acc @1(↑)* | *Acc @5(↑)* | | *Acc @1(↑)* | *Acc @5(↑)* | | *Acc @1(↑)* | *Acc @5(↑)* | |
| | Baseline | 91.55 ± 0.01 | 99.76 ± 0.01 | 246716.268 | 69.96 ± 0.56 | 91.64 ± 0.25 | 246738.386 | 66.38 ± 0.96 | 86.22 ± 1.05 | 493525.924 |
| | Pe = 5 | 90.87 ± 0.25 | 99.71 ± 0.06 | 206008.083 | 68.02 ± 0.86 | 90.39 ± 0.20 | 206026.552 | 64.90 ± 0.71 | 85.39 ± 0.30 | 412094.146 |
| | Pe = 10 | **91.66 ± 0.18** | **99.86 ± 0.04** | 225745.385 | **69.98 ± 0.33** | **91.86 ± 0.04** | 225765.623 | **66.40 ± 0.85** | **86.89 ± 0.16** | 451576.22 |
| | Pr = 0.2 | **91.93 ± 0.08** | **99.79 ± 0.06** | 217110.315 | 68.42 ± 1.34 | **91.77 ± 0.42** | 217129.78 | **66.89 ± 1.37** | **86.70 ± 0.76** | 434302.813 |
| | Pr = 0.5 | 89.94 ± 0.31 | 99.69 ± 0.01 | 166533.481 | 65.90 ± 1.36 | 89.71 ± 0.49 | 166548.41 | 61.59 ± 1.67 | 83.82 ± 1.44 | 333129.999 |
| ResNet-50 | Pr = 0.7 | 89.28 ± 0.06 | 99.59 ± 0.03 | 146796.179 | 64.21 ± 1.07 | 89.05 ± 0.29 | 146809.34 | 60.64 ± 1.06 | 83.11 ± 1.37 | 293647.925 |
| | DP = 5 | 91.32 ± 0.23 | 99.75 ± 0.05 | 234380.454 | 69.53 ± 0.64 | 90.90 ± 0.21 | 234401.467 | 65.20 ± 0.28 | 86.22 ± 0.06 | 468849.628 |
| | DP = 10 | 91.61 ± 0.08 | 99.78 ± 0.02 | 240548.361 | **69.98 ± 0.25** | 91.70 ± 0.20 | 240569.926 | **66.63 ± 0.47** | 86.27 ± 0.33 | 481187.776 |
| | DR = 0.2 | 91.23 ± 0.21 | **99.8 ± 0.03** | 234380.454 | **70.06 ± 0.04** | 91.73 ± 0.05 | 234401.467 | 66.39 ± 0.11 | 86.39 ± 0.38 | 468849.628 |
| | DR = 0.5 | 90.86 ± 0.33 | 99.69 ± 0.03 | 215876.734 | 68.18 ± 0.00 | 90.81 ± 0.01 | 215896.088 | 64.82 ± 0.40 | 85.52 ± 0.48 | 431835.183 |
| | DR = 0.7 | 91.06 ± 0.06 | 99.70 ± 0.02 | 203540.921 | 68.27 ± 0.14 | 90.73 ± 0.04 | 203559.168 | 64.14 ± 0.25 | 85.77 ± 0.18 | 407158.887 |
| | Baseline | 86.27 ± 1.05 | 99.33 ± 0.18 | 261786.175 | 64.53 ± 1.54 | 88.58 ± 1.92 | 261790.322 | 62.74 ± 1.18 | 84.69 ± 1.14 | 523589.861 |
| | Pe = 5 | 84.39 ± 1.11 | 99.22 ± 0.11 | 218591.456 | 61.95 ± 1.83 | 87.05 ± 1.44 | 218594.919 | 60.80 ± 1.98 | 83.28 ± 1.48 | 437197.534 |
| | Pe = 10 | **86.36 ± 1.41** | **99.36 ± 0.12** | 239534.35 | **64.86 ± 1.54** | **88.66 ± 1.05** | 239538.145 | **62.75 ± 1.16** | **84.99 ± 1.15** | 479084.723 |
| | Pr = 0.2 | **86.81 ± 1.67** | **99.35 ± 0.15** | 230371.834 | **64.58 ± 1.01** | **88.59 ± 1.62** | 230375.484 | **62.83 ± 1.22** | **84.99 ± 1.48** | 460759.078 |
| | Pr = 0.5 | 82.17 ± 1.18 | 98.97 ± 0.24 | 176705.668 | 57.99 ± 1.76 | 84.96 ± 1.64 | 176708.468 | 58.53 ± 1.59 | 81.63 ± 1.71 | 353423.156 |
| Swin-T | Pr = 0.7 | 80.45 ± 1.73 | 98.95 ± 0.06 | 155762.774 | 55.41 ± 1.03 | 83.25 ± 4.05 | 155765.242 | 57.58 ± 1.63 | 80.94 ± 1.81 | 311535.967 |
| | DP = 5 | 84.61 ± 0.50 | 99.20 ± 0.09 | 248696.866 | 61.13 ± 0.37 | 86.72 ± 0.23 | 248700.806 | 61.62 ± 0.31 | 83.82 ± 0.17 | 497410.368 |
| | DP = 10 | **86.92 ± 0.27** | **99.37 ± 0.12** | 255241.521 | **64.57 ± 0.19** | **88.96 ± 0.16** | 255245.564 | **62.75 ± 0.06** | **85 ± 0.07** | 510500.114 |
| | DR = 0.2 | **86.56 ± 0.57** | **99.38 ± 0.02** | 248696.866 | **64.99 ± 0.06** | **88.78 ± 0.36** | 248700.806 | **62.77 ± 0.49** | **84.99 ± 0.44** | 497410.368 |
| | DR = 0.5 | 83.96 ± 0.03 | 99.12 ± 0.08 | 229062.903 | 60.39 ± 0.22 | 85.95 ± 0.28 | 229066.532 | 61.19 ± 0.27 | 83.79 ± 0.04 | 458141.128 |
| | DR = 0.7 | 83.63 ± 0.29 | 99.13 ± 0.06 | 215973.595 | 59.60 ± 0.04 | 85.82 ± 0.25 | 215977.016 | 60.59 ± 0.16 | 83.37 ± 0.38 | 431961.635 |
| | Baseline | 81.99 ± 1.32 | 98.95 ± 0.23 | 155631.206 | 58.04 ± 1.43 | 83.71 ± 1.44 | 155633.971 | 54.02 ± 1.59 | 77.88 ± 1.35 | 311274.086 |
| | Pe = 5 | 80.61 ± 1.21 | 98.93 ± 0.11 | 129952.057 | 55.23 ± 0.92 | 82.11 ± 0.91 | 129954.366 | 51.71 ± 1.23 | 76.22 ± 1.43 | 259913.862 |
| | Pe = 10 | **82.20 ± 0.83** | **98.96 ± 0.15** | 142402.554 | **58.31 ± 1.36** | **83.80 ± 1.05** | 142405.084 | **54.85 ± 1.50** | **77.85 ± 1.39** | 284815.789 |
| | Pr = 0.2 | **82 ± 1.44** | **98.99 ± 0.13** | 136955.462 | **58.20 ± 1.04** | **83.73 ± 1.20** | 136957.895 | **54.65 ± 1.43** | 77.86 ± 1.49 | 273921.196 |
| | Pr = 0.5 | 78.96 ± 1.29 | 98.66 ± 0.25 | 105051.064 | 52.71 ± 1.92 | 80.73 ± 1.39 | 105052.931 | 48.97 ± 1.43 | 73.96 ± 1.22 | 210110.008 |
| MLP-Mixer | Pr = 0.7 | 77.31 ± 1.36 | 98.55 ± 0.21 | 92600.568 | 50.76 ± 1.17 | 79.54 ± 0.76 | 92602.213 | 46.52 ± 1.48 | 72.66 ± 1.33 | 185208.081 |
| | DP = 5 | 80.49 ± 1.21 | 98.82 ± 0.03 | 147849.646 | 56.09 ± 0.42 | 82.68 ± 0.04 | 147852.273 | 51.99 ± 1.50 | 76.36 ± 1.44 | 295710.382 |
| | DP = 10 | **82.05 ± 0.93** | **98.97 ± 0.04** | 151740.426 | **58.36 ± 0.57** | **83.73 ± 0.17** | 151743.122 | **54.96 ± 1.50** | **77.86 ± 1.28** | 303492.234 |
| | DR = 0.2 | 81.22 ± 0.85 | **98.96 ± 0.07** | 147849.646 | **58.14 ± 0.57** | **83.75 ± 0.04** | 147852.273 | **54.87 ± 1.43** | **77.89 ± 1.26** | 295710.382 |
| | DR = 0.5 | 80.27 ± 1.13 | 98.78 ± 0.06 | 136177.306 | 55.24 ± 0.28 | 82.34 ± 0.00 | 147852.273 | 51.00 ± 1.37 | 75.94 ± 1.22 | 272364.826 |
| | DR = 0.7 | 80.00 ± 1.18 | 98.79 ± 0.03 | 128395.745 | 55.08 ± 0.76 | 82.10 ± 0.12 | 128398.026 | 50.78 ± 1.49 | 75.68 ± 1.35 | 256801.121 |

Table 3: Performance comparison of standard models upon ImageNet dataset under diverse sampling strategies, including periodic (Pe), probabilistic (Pr) samplings over 100 epochs for 3 independent trials. Best values are **bolded**.

| Strategy | ResNet-50 | | Swin-T | | MLP-Mixer | |
|---|---|---|---|---|---|---|
| | *Acc @1(↑)* | *Acc @5(↑)* | *Acc @1(↑)* | *Acc @5(↑)* | *Acc @1(↑)* | *Acc @5(↑)* |
| Baseline | 57.75±1.23 | 77.25±0.89 | 55.45±1.05 | 75.15±0.98 | 47.32±1.40 | 65.45±0.95 |
| Pe = 5 | 56.85±0.94 | 77.15±1.67 | 55.25±0.79 | 74.95±1.53 | 45.75±0.85 | 64.25±1.42 |
| Pe = 10 | **57.82±1.45** | **78.65±0.76** | **55.47±1.31** | **76.75±1.08** | **48.07±1.67** | **65.75±0.83** |
| Pr = 0.2 | **58.05±0.92** | **78.20±1.31** | **56.07±1.45** | **76.20±0.72** | **48.01±1.15** | **65.65±1.31** |
| Pr = 0.5 | 55.08±1.56 | 76.65±0.84 | 54.38±0.91 | 74.65±1.60 | 45.25±0.78 | 63.25±1.49 |
| Pr = 0.7 | 56.00±0.74 | 76.85±1.42 | 53.07±1.22 | 73.85±1.35 | 46.09±1.53 | 63.75±0.92 |

By examining the results from the table, RT-DETRv2 (Lv et al., 2024) consistently outperforms the Baseline in both $mAP$ @.5 and $mAP$ @.5 : .95 for all sampling strategies, except at Pr = 0.2 for $mAP$ @.5, where it matches the Baseline. For YOLOv7 (Wang et al., 2023), sampling strategies Pe = 10 and Pr = 0.2 demonstrate better performance than the Baseline while achieving energy reductions of 10% and 20%, respectively. These reductions result from skipping epochs, thereby lowering the total effective TFLOPS.

Table 4: Performance comparison of various sampling strategies on detection models, **YOLOv7** and **RT-DETRv2**, evaluated on the PascalVOC-2012 dataset over 3 independent trials. Best results are highlighted in **bold**.

| Strategy | YOLOv7 | | RT-DETRv2 | |
|---|---|---|---|---|
| | $mAP$ @.5 ($\uparrow$) | $mAP$ @.5 : .95 ($\uparrow$) | $mAP$ @.5 ($\uparrow$) | $mAP$ @.5 : .95 ($\uparrow$) |
| Baseline | $77.1 \pm 0.73$ | $59.6 \pm 1.22$ | $81.6 \pm 1.35$ | $66.7 \pm 0.88$ |
| Pe = 5 | $76.0 \pm 1.03$ | $58.5 \pm 0.84$ | $\mathbf{81.9 \pm 1.15}$ | $\mathbf{67.1 \pm 0.72}$ |
| Pe = 10 | $\mathbf{77.4 \pm 0.95}$ | $\mathbf{59.7 \pm 1.04}$ | $\mathbf{82.0 \pm 1.19}$ | $\mathbf{67.4 \pm 0.81}$ |
| Pr = 0.2 | $\mathbf{77.2 \pm 0.82}$ | $\mathbf{59.9 \pm 0.93}$ | $81.6 \pm 1.21$ | $\mathbf{66.9 \pm 0.94}$ |
| Pr = 0.5 | $75.9 \pm 1.04$ | $58.2 \pm 0.52$ | $\mathbf{82.0 \pm 0.84}$ | $\mathbf{67.1 \pm 0.92}$ |
| Pr = 0.7 | $74.8 \pm 0.75$ | $57.1 \pm 1.02$ | $\mathbf{82.2 \pm 1.07}$ | $\mathbf{67.1 \pm 0.64}$ |

Table 5: Performance comparison of different sampling strategies on segmentation models, **U-Net** and **Segmenter**, evaluated on the ADE20K dataset over 3 independent trials. Best results are highlighted in **bold**.

| Strategy | U-Net | | Segmenter | |
|---|---|---|---|---|
| | $SS\text{-}IoU$ ($\uparrow$) | $MS\text{-}IoU$ ($\uparrow$) | $SS\text{-}IoU$ ($\uparrow$) | $MS\text{-}IoU$ ($\uparrow$) |
| Baseline | $27.50 \pm 1.34$ | $28.42 \pm 1.15$ | $38.02 \pm 1.52$ | $38.78 \pm 1.23$ |
| Pe = 5 | $26.53 \pm 1.73$ | $27.31 \pm 1.65$ | $37.94 \pm 1.14$ | $\mathbf{38.84 \pm 0.98}$ |
| Pe = 10 | $\mathbf{27.53 \pm 1.91}$ | $\mathbf{28.52 \pm 1.04}$ | $\mathbf{38.15 \pm 1.46}$ | $\mathbf{38.91 \pm 1.32}$ |
| Pr = 0.2 | $\mathbf{27.75 \pm 1.54}$ | $\mathbf{28.50 \pm 1.27}$ | $\mathbf{38.16 \pm 0.83}$ | $\mathbf{38.79 \pm 1.64}$ |
| Pr = 0.5 | $25.13 \pm 0.91$ | $25.73 \pm 1.42$ | $36.26 \pm 0.78$ | $38.24 \pm 1.12$ |
| Pr = 0.7 | $23.91 \pm 0.67$ | $24.49 \pm 0.84$ | $36.16 \pm 1.05$ | $37.37 \pm 1.82$ |

**Image Segmentation:** For the image segmentation task, we evaluated the performance of the widely used U-Net (Ronneberger et al., 2015) and the transformer-based Segmenter (Strudel et al., 2021) models, as presented in Table 5. The results demonstrate that sampling strategies such as Pe = 10 and Pr = 0.2 consistently outperform the baseline in terms of both *SS-IoU* and *MS-IoU*. These strategies enable significant energy savings of 10% and 20%, respectively. The observed reduction in energy consumption is primarily attributed to the skipped epochs during training, which directly decrease the effective TFLOPS required. This finding highlights the potential of incorporating such sampling strategies to balance model performance with computational efficiency, making them particularly beneficial in resource-constrained scenarios.

**Domain Adaptation (DA) and Domain Generalization (DG):** Domain Adaptation (DA) focuses on transferring knowledge from a labeled source domain to an unlabeled target domain, addressing distribution shifts. Domain Generalization (DG), on the other hand, aims to learn models that perform well on unseen target domains without accessing their data during training. We experimented with some of DA strategies like Marginal Disparity Descrepency (MDD) and Minimum Class Confusion (MCC) upon Office-31 (Saenko et al., 2010) dataset. While for DG we experimentedd with VREx (Krueger et al., 2021) and GroupDRO (Sagawa et al., 2020) upon PACS (Li et al., 2017) as shown in the tables 6 and 7

From the results in tables 6 and 7, it is evident that most of the sampling strategies for **MDD** (Zhang et al., 2019), **MCC** (Jin et al., 2020), **Vrex** (Krueger et al., 2021), and **GroupDRO** (Sagawa et al., 2020) outperform the Baseline models. In particular, the strategies with Pe = 10 and Pr = 0.2 stand out, yielding substantial improvements that result in energy savings of up to 10% and 20%. This underscores the effectiveness of our sampling strategy in promoting energy efficiency, even in the context of domain shifts, where models must adapt to an unknown target domain.

**Federated Learning (FL):** We evaluated the FL agorithms (McMahan et al., 2017; Li et al., 2020b; Sun et al., 2023) upon different sampling strategies and summarized in table 8 for both IID and Non-IID cases.

Table 6: Performance comparision of different sampling strategies on DA methods, **MDD** and **MCC**, evaluated upon Office-31 dataset over 3 independent trials. Best results are highlighted in **bold**.

| Method | Strategy | A-W | D-W | W-D | A-D | D-A | W-A |
|---|---|---|---|---|---|---|---|
| | Baseline | 92.15 ± 0.89 | 98.53 ± 0.07 | 99.93 ± 0.12 | 91.96 ± 0.88 | 74.81 ± 0.87 | 72.25 ± 1.15 |
| | Pe = 5 | **92.91 ± 0.14** | **98.57 ± 0.19** | **100.00 ± 0.00** | **92.03 ± 1.21** | **74.94 ± 0.84** | 71.95 ± 0.87 |
| MDD | Pe = 10 | 92.74 ± 0.88 | 98.49 ± 0.13 | 99.93 ± 0.12 | **93.37 ± 0.60** | 74.84 ± 0.95 | 72.44 ± 0.87 |
| | Pr = 0.2 | 91.65 ± 0.85 | 98.53 ± 0.19 | 99.93 ± 0.12 | 92.63 ± 1.31 | 74.90 ± 0.88 | 72.39 ± 0.35 |
| | Pr = 0.5 | 91.99 ± 1.58 | **98.78 ± 0.19** | **100.00 ± 0.00** | 92.83 ± 1.82 | 73.68 ± 0.84 | 71.47 ± 0.06 |
| | Pr = 0.7 | 91.15 ± 0.19 | **98.90 ± 0.19** | **100.00 ± 0.00** | 91.83 ± 0.65 | 73.42 ± 1.15 | 71.00 ± 1.67 |
| | Baseline | 93.29 ± 0.19 | 98.27 ± 0.08 | 99.72 ± 0.12 | 93.64 ± 0.46 | 75.24 ± 0.86 | 74.54 ± 0.94 |
| | Pe = 5 | 93.38 ± 0.28 | 98.15 ± 0.14 | 99.72 ± 0.12 | **93.84 ± 0.64** | **75.49 ± 0.73** | 74.52 ± 0.82 |
| MCC | Pe = 10 | 93.24 ± 0.36 | **98.36 ± 0.08** | 99.66 ± 0.12 | 93.67 ± 1.06 | 75.37 ± 0.37 | 74.57 ± 0.62 |
| | Pr = 0.2 | **93.50 ± 0.14** | **98.53 ± 0.13** | 99.72 ± 0.12 | **94.17 ± 0.40** | **75.52 ± 0.62** | **74.6 ± 0.85** |
| | Pr = 0.5 | 93.24 ± 0.08 | 98.31 ± 0.07 | 99.72 ± 0.12 | 93.57 ± 0.53 | 75.26 ± 0.56 | 74.37 ± 0.80 |
| | Pr = 0.7 | **93.55 ± 0.33** | 98.27 ± 0.08 | 99.72 ± 0.12 | 93.73 ± 1.29 | 75.27 ± 0.38 | **74.58 ± 0.48** |

Table 7: Performance comparision of different sampling strategies on DG methods, **Vrex** and **GroupDRO**, evaluated upon PACS dataset over 3 independent trials. Best results are highlighted in **bold**.

| Method | Strategy | P | A | C | S |
|---|---|---|---|---|---|
| | Baseline | 97.66 ± 0.18 | 87.84 ± 0.69 | 79.45 ± 1.08 | 80.72 ± 2.05 |
| | Pe = 5 | 97.44 ± 0.38 | **88.36 ± 1.93** | 79.19 ± 0.28 | 80.39 ± 0.62 |
| Vrex | Pe = 10 | **97.76 ± 0.09** | 88.34 ± 0.59 | **80.90 ± 1.71** | **81.65 ± 1.11** |
| | Pr = 0.2 | **97.76 ± 0.28** | 87.71 ± 1.88 | **80.23 ± 1.52** | **81.89 ± 0.38** |
| | Pr = 0.5 | 97.52 ± 0.28 | **87.98 ± 0.32** | **80.69 ± 0.39** | 81.38 ± 2.51 |
| | Pr = 0.7 | 97.05 ± 0.27 | **88.34 ± 1.25** | 80.02 ± 2.15 | 81.82 ± 2.12 |
| | Baseline | 97.92 ± 0.15 | 88.60 ± 0.23 | 80.46 ± 1.90 | 79.32 ± 2.32 |
| | Pe = 5 | 97.70 ± 0.03 | **88.96 ± 1.01** | **81.05 ± 0.39** | **79.74 ± 2.59** |
| GroupDRO | Pe = 10 | **97.98 ± 0.33** | **88.98 ± 0.16** | **80.96 ± 1.86** | 79.36 ± 1.58 |
| | Pr = 0.2 | **97.99 ± 0.27** | **88.99 ± 0.93** | **80.90 ± 1.33** | **79.90 ± 0.28** |
| | Pr = 0.5 | 97.66 ± 0.16 | 88.42 ± 1.14 | 80.09 ± 1.59 | 78.95 ± 0.96 |
| | Pr = 0.7 | 97.50 ± 0.33 | **88.99 ± 1.74** | 79.96 ± 1.75 | 78.90 ± 1.77 |

For the Non-IID split, we use the `Dirichlet` distribution (Hsu et al., 2019) with a parameter of 0.6. Upon

Table 8: Performance comparision of different sampling strategies upon FL methods like `FedAvg`, `FedProx` and `FedSpeed`, evaluated upon CIFAR-10 dataset over 3 independent trails. Best results are highlighted in **bold.**

| Strategy | FedAvg | | FedProx | | FedSpeed | |
|---|---|---|---|---|---|---|
| | *IID* | *Non-IID (Dir 0.6)* | *IID* | *Non-IID (Dir 0.6)* | *IID* | *Non-IID (Dir 0.6)* |
| Baseline | 82.62 ± 1.05 | 80.80 ± 0.97 | 79.40 ± 1.34 | 77.51 ± 1.42 | 87.47 ± 0.82 | 86.52 ± 0.94 |
| Pe = 5 | 82.35 ± 1.13 | **80.82 ± 1.02** | **79.54 ± 0.89** | **78.18 ± 1.01** | 87.43 ± 1.10 | 86.01 ± 0.78 |
| Pe = 10 | 82.71 ± 0.91 | **81.01 ± 1.34** | 79.55 ± 1.26 | 77.85 ± 1.10 | 87.53 ± 0.88 | **86.58 ± 1.33** |
| Pr = 0.2 | **82.72 ± 0.88** | 80.91 ± 1.26 | 79.51 ± 1.03 | 77.44 ± 0.98 | 87.49 ± 1.29 | **86.56 ± 1.16** |
| Pr = 0.5 | 81.81 ± 1.14 | 79.67 ± 0.93 | 79.20 ± 0.94 | **77.79 ± 1.16** | 85.79 ± 0.77 | 85.79 ± 1.48 |
| Pr = 0.7 | 81.75 ± 0.76 | 80.08 ± 1.27 | 79.34 ± 1.05 | **77.97 ± 1.37** | 85.04 ± 1.12 | 85.04 ± 0.96 |

inspecting Table 8, most federated learning (FL) algorithms outperform the baseline, with Pe = 10 and Pr = 0.2 achieving 10% and 20% energy savings, respectively. These gains are particularly valuable in FL, where multiple local epochs reduce energy consumption and FLOPs, enhancing computational efficiency.

Thus, strategies like Pe = 10 and Pr = 0.2 strike a balance between performance, energy savings, and efficiency, improving scalability and practicality for resource-constrained, real-world FL applications.

**LLM Pre-training:** We evaluated our sampling strategies on GPT-2 (Radford et al., 2019) fine-tuned over the IMDB dataset (Maas et al., 2011) for 20 epochs using the Adam optimizer (learning rate = 0.02). The results 9 demonstrate consistent improvements across multiple metrics. Importantly, these improvements are achieved alongside substantial energy savings—up to 70% in some cases—demonstrating the practicality of our sampling strategy for efficient training of large language models, where computational cost is a critical concern.

Table 9: Performance comparison of different sampling strategies applied to GPT-2 (Radford et al., 2019) fine-tuned on the IMDB (Maas et al., 2011) dataset, evaluated over 3 independent trials. Best results are highlighted in **bold**.

| Strategy | Accuracy (↑) | Precision (↑) | Recall (↑) | F1-score (↑) | Avg |
|---|---|---|---|---|---|
| Baseline | 55.74 | 92.96 | 12.65 | 22.27 | 45.91 |
| Pe = 5 | 55.68 | 87.56 | **13.49** | **23.37** | 45.53 |
| Pe = 10 | **55.88** | 93.12 | 13.05 | 22.95 | 46.25 |
| Pr = 0.2 | **56.02** | 92.75 | 13.20 | 23.10 | 46.27 |
| Pr = 0.5 | **56.14** | 89.22 | 14.21 | 24.51 | 46.02 |
| Pr = 0.7 | **55.86** | 89.03 | 13.61 | 23.61 | 45.53 |

**Optimizers:** We tested our sampling approaches on various optimizers, including AdaGrad (Duchi et al., 2011) and AdaDelta (Zeiler, 2012), to assess their impact on energy savings. A careful analysis of the results in Table 10 demonstrates that our sampling methods can be successfully applied to different optimizers without sacrificing performance. Notably, in the cases of Pe = 10 and Pr = 0.2, the energy savings achieved were 10% and 20%, respectively.

Table 10: Impact of Different Optimizers (Ada-Grad vs. Ada-Delta) on Accuracy (%). Mean ± Std are reported for each strategy.

| Strategy | Ada-Grad ($Acc@1$ (%), $Acc@5$ (%)) | Ada-Delta ($Acc@1$ (%), $Acc@5$ (%)) |
|---|---|---|
| Baseline | $79.90 \pm 1.25$ , $94.69 \pm 1.84$ | $88.90 \pm 1.14$ , $99.66 \pm 2.13$ |
| Pe = 5 | $79.50 \pm 1.15$ , $93.50 \pm 2.64$ | $\mathbf{88.96 \pm 1.15}$ , $99.66 \pm 2.27$ |
| Pe = 10 | $\mathbf{79.95 \pm 1.30}$ , $\mathbf{94.75 \pm 2.12}$ | $\mathbf{89.92 \pm 1.20}$ , $\mathbf{99.68 \pm 2.03}$ |
| Pr = 0.2 | $\mathbf{79.97 \pm 1.10}$ , $\mathbf{94.79 \pm 1.95}$ | $88.90 \pm 1.05$ , $\mathbf{99.68 \pm 2.50}$ |
| Pr = 0.5 | $77.73 \pm 1.20$ , $93.64 \pm 2.70$ | $87.52 \pm 1.30$ , $99.46 \pm 2.18$ |
| Pr = 0.7 | $75.45 \pm 1.35$ , $93.68 \pm 1.67$ | $87.04 \pm 1.40$ , $99.51 \pm 2.38$ |

**Comparison with Related Works:** We experimented with the recent MeZO Malladi et al. (2023) method on the SST-2 Socher et al. (2013) and RTE Dagan et al. (2005) datasets using 10k optimization steps. The results, along with the performance of our proposed strategies, are reported in Table 11.

Table 11: Performance comparison of different strategies on SST-2 and RTE datasets.

| | Strategy | SST-2 | RTE |
|---|---|---|---|
| | Baseline | 91.28 | 67.57 |
| | Pe = 5 | 90.67 | 65.78 |
| MeZO | Pe = 10 | **91.67** | **67.85** |
| | Pr = 0.2 | **92.07** | **67.75** |
| | Pr = 0.5 | **91.36** | 67.55 |
| | Pr = 0.7 | 90.97 | 67.02 |

As shown in Table 11, our proposed strategies consistently outperform the baseline MeZO method in terms of accuracy on both SST-2 and RTE datasets.Notably, the sampling conditions with Pe = 10 and Pr = 0.2 yield improved performance compared to the baseline approach. This demonstrates the scalability and effectiveness of our method across diverse settings without compromising the performance.

## 5    Conclusion

We introduced a gradient sampling technique to reduce energy consumption in deep learning models, particularly for computer vision tasks such as image classification, object detection, and image segmentation. Our approach also demonstrated effectiveness in out-of-distribution scenarios, including Domain Adaptation (DA), Domain Generalization (DG), and Federated Learning (FL). We further validated its applicability in large language model (LLM) pretraining, where it achieved similar energy savings without degrading model quality. Tested across various optimizers, it proved to be robust and practical, showing that periodic and probabilistic sampling can significantly reduce energy usage without compromising performance.

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

# A Theorem 3.1 Proof:

Suppose $\hat{\boldsymbol{\delta}}^{(k)} \sim \text{GGD}((\boldsymbol{\theta}^{(k)} - \boldsymbol{\theta}^{(k-1)}); \mu^{(k)}, \sigma^{(k)}, \beta^{(k)})$, where GGD denotes the Generalized Gaussian Distribution. Then, the parameter update at the $(k+1)$-th epoch, when sampling via SGD, is given by:

$$\boldsymbol{\theta}^{(k+1)} = \begin{cases} \boldsymbol{\theta}^{(k)} + \hat{\boldsymbol{\delta}}^{(k)} & \text{with probability } \rho \\ \boldsymbol{\theta}^{(k)} - \eta \nabla \mathcal{L}(\boldsymbol{\theta}^{(k)}) & \text{with probability } (1 - \rho) \end{cases}$$

The expected value of the loss at the next iteration, $\mathbb{E}[\mathcal{L}(\boldsymbol{\theta}^{(k+1)})]$, can be written as:

$$\mathbb{E}[\mathcal{L}(\boldsymbol{\theta}^{(k+1)})] = \rho \left[ \underbrace{\mathbb{E}[\mathcal{L}(\boldsymbol{\theta}^{(k+1)})|\text{GGD}]}_{\text{Lemma A.1}} \right] + (1 - \rho) \left[ \underbrace{\mathbb{E}[\mathcal{L}(\boldsymbol{\theta}^{(k+1)})|\text{SGD}]}_{\text{Lemma A.2}} \right] \tag{13}$$

**Lemma A.1.** *Assume that the function $\mathcal{L}(\boldsymbol{\theta}^{(k)})$ is $L'$-Lipschitz continuous, with $\mathbb{E}[\hat{\boldsymbol{\delta}}^{(k)}] = \mu^{(k)} = 0$ and $\mathbb{E}[||\hat{\boldsymbol{\delta}}^{(k)}||^2] \leq \sigma^{(k)^2}$ (bounded). Then,*

$$\mathbb{E}[\mathcal{L}(\boldsymbol{\theta}^{(k+1)})] \leq \mathbb{E}[\mathcal{L}(\boldsymbol{\theta}^{(k)})] + \frac{L'\sigma^{(k)^2}}{2}.$$

*Proof.* Since $\mathcal{L}(.)$ is $L'$-Lipschitz continuous, with $\mathbb{E}[\hat{\boldsymbol{\delta}}^{(k)}] = \mu^{(k)} = 0$, $\quad \mathbb{E}[||\hat{\boldsymbol{\delta}}^{(k)}||^2] \leq \sigma^{(k)^2}$, we have:

$$\mathcal{L}(\boldsymbol{\theta}^{(k+1)}) \leq \mathcal{L}(\boldsymbol{\theta}^{(k)}) + \nabla \mathcal{L}(\boldsymbol{\theta}^{(k)})^\top \hat{\boldsymbol{\delta}}^{(k)} + \frac{L'}{2} ||\hat{\boldsymbol{\delta}}^{(k)}||^2$$

$$\mathbb{E}[\mathcal{L}(\boldsymbol{\theta}^{(k+1)})] \leq \mathbb{E}[\mathcal{L}(\boldsymbol{\theta}^{(k)})] + \frac{L'\sigma^{(k)^2}}{2}. \tag{14}$$

$\square$

**Lemma A.2.** *Assume that the function $\mathcal{L}(\boldsymbol{\theta}^{(k)})$ is both $L'$-Lipschitz continuous and $\mu'$-strongly convex. Then $\boldsymbol{\theta}^{(k+1)}$ follows the SGD update rule:*

$$\mathbb{E}[\mathcal{L}(\boldsymbol{\theta}^{(k+1)})] \leq \mathbb{E}[\mathcal{L}(\boldsymbol{\theta}^{(*)})] + (1 - 2\mu'\eta)\Delta_k,$$

*where $\Delta_k = \mathbb{E}[\mathcal{L}(\boldsymbol{\theta}^{(k)}) - \mathcal{L}(\boldsymbol{\theta}^{(*)})]$ and $\boldsymbol{\theta}^{(*)}$ is the optimal parameter.*

*Proof.* Since $\mathcal{L}(.)$ is $L'$-Lipschitz continuous, we have:

$$\mathcal{L}(\boldsymbol{\theta}^{(k+1)}) \leq \mathcal{L}(\boldsymbol{\theta}^{(k)}) + \nabla \mathcal{L}(\boldsymbol{\theta}^{(k)})^\top (-\eta \nabla \mathcal{L}(\boldsymbol{\theta}^{(k)})) + \frac{L'}{2} || - \eta \nabla \mathcal{L}(\boldsymbol{\theta}^{(k)})||^2$$

$$\leq \mathcal{L}(\boldsymbol{\theta}^{(k)}) - \left( \eta - \frac{L'}{2}\eta^2 \right) ||\nabla \mathcal{L}(\boldsymbol{\theta}^{(k)})||^2 \tag{15}$$

Since $\mathcal{L}(.)$ is $\mu'$-strongly convex, we have the inequality:

$$||\nabla \mathcal{L}(\boldsymbol{\theta}^{(k)})||^2 \geq 2\mu'(\mathcal{L}(\boldsymbol{\theta}^{(k)}) - \mathcal{L}(\boldsymbol{\theta}^{(*)})) \tag{16}$$

Substituting equation 16 into equation 15, we get:

$$\mathcal{L}(\boldsymbol{\theta}^{(k+1)}) \leq \mathcal{L}(\boldsymbol{\theta}^{(k)}) - 2\mu' \left( \eta - \frac{L'}{2}\eta^2 \right) (\mathcal{L}(\boldsymbol{\theta}^{(k)}) - \mathcal{L}(\boldsymbol{\theta}^{(*)}))$$

$$\mathcal{L}(\boldsymbol{\theta}^{(k+1)}) \leq \mathcal{L}(\boldsymbol{\theta}^{(*)}) + (1 - 2\mu'\eta + L'\mu'\eta^2)(\mathcal{L}(\boldsymbol{\theta}^{(k)}) - \mathcal{L}(\boldsymbol{\theta}^{(*)}))$$

Taking $\mathbb{E}[\cdot]$ on both sides

$$\mathbb{E}[\mathcal{L}(\boldsymbol{\theta}^{(k+1)})] \leq \mathbb{E}[\mathcal{L}(\boldsymbol{\theta}^{(*)})] + (1 - 2\mu'\eta + L'\mu'\eta^2)\Delta_k \quad \left( \Delta_k = \mathbb{E}[\mathcal{L}(\boldsymbol{\theta}^{(k)}) - \mathcal{L}(\boldsymbol{\theta}^{(*)})] \right) \tag{17}$$

$\square$

Combining equation 14 and equation 17, we can rewrite equation equation 13 as:

$$\mathbb{E}[\mathcal{L}(\boldsymbol{\theta}^{(k+1)})] \leq \rho \left[ \mathbb{E}[\mathcal{L}(\boldsymbol{\theta}^{(k)})] + \frac{L'\sigma^{(k)^2}}{2} \right] + (1-\rho) \left[ \mathbb{E}[\mathcal{L}(\boldsymbol{\theta}^{(*)})] + (1-2\mu'\eta + L'\mu'\eta^2)\Delta_k \right].$$

Rearranging this, we obtain the following recurrence relation for the loss difference:

$$\mathbb{E}[\mathcal{L}(\boldsymbol{\theta}^{(k+1)})] \leq \rho \left( \underbrace{\mathbb{E}[\mathcal{L}(\boldsymbol{\theta}^{(k)}) - \mathcal{L}(\boldsymbol{\theta}^{(*)})]}_{\Delta_k} \right) + \frac{\rho L'\sigma^{(k)^2}}{2} + \mathbb{E}[\mathcal{L}(\boldsymbol{\theta}^{(*)})] + (1-\rho)(1-2\mu'\eta + L'\mu'\eta^2)\Delta_k$$

$$\Delta_{k+1} \leq \rho\Delta_k + (1-\rho)(1-2\mu'\eta + L'\mu'\eta^2)\Delta_k + \frac{\rho L'\sigma^{(k)^2}}{2}$$

$$\Delta_{k+1} \leq \left( \underbrace{(1-(1-\rho)(2\mu'\eta - L'\mu'\eta^2)}_{\gamma} \right) \Delta_k + \frac{\rho L'\sigma^{(k)^2}}{2} \tag{18}$$

Now, we analyze the recurrence relation for the loss difference:

$$\Delta_k = \mathbb{E}[\mathcal{L}(\boldsymbol{\theta}^{(k)}) - \mathcal{L}(\boldsymbol{\theta}^{(*)})].$$

At steady state ($k \to \infty$), we assume that $\Delta_{k+1} = \Delta_k = \Delta_*$. Substituting this into the recurrence relation, we get:

$$\Delta_* = \gamma\Delta_* + \frac{\rho L'\sigma^{(k)^2}}{2}.$$

Rearranging the terms:

$$\Delta_*(1-\gamma) = \frac{\rho L'\sigma^{(k)^2}}{2},$$

which simplifies to:

$$\Delta_* = \frac{\rho L'\sigma^{(k)^2}}{2(1-\gamma)}.$$

Substituting $\gamma = 1 - (1-\rho)(2\mu'\eta - L'\mu'\eta^2)$, we get:

$$\Delta_* = \frac{\rho L'\sigma^{(k)^2}}{2(1-\rho)(2\mu'\eta - L'\mu'\eta^2)}.$$

Next, we consider the transient behavior. The recurrence relation for the loss difference is:

$$\Delta_{k+1} = \gamma\Delta_k + \frac{\rho L'\sigma^{(k)^2}}{2}.$$

The general solution to this recurrence is:

$$\Delta_k = (\Delta_0 - \Delta_*)\gamma^k + \Delta_*,$$

where $\Delta_0 = \mathbb{E}[\mathcal{L}(\boldsymbol{\theta}^{(0)}) - \mathcal{L}(\boldsymbol{\theta}^{(*)})]$ is the initial loss difference. The convergence rate of the recurrence is determined by the term $(\Delta_0 - \Delta_*)\gamma^k$, which decays exponentially as $k$ increases. Since $0 < \gamma < 1$, the rate of decay is controlled by $\gamma = 1 - (1 - \rho)(2\mu'\eta - L'\mu'\eta^2)$. For convergence to occur, we require $0 < \gamma < 1$.

The steady-state value $\Delta_*$ is proportional to:

$$\Delta_* \propto \frac{\rho L' \sigma^{(k)^2}}{(1 - \rho)(2\mu'\eta - L'\mu'\eta^2)}.$$

This implies that increasing $\mu'$ (the strong convexity constant) or $\eta$ (the learning rate) reduces the steady-state error $\Delta_*$. The steady-state error also depends on the probability $\rho$, which governs the sampling process. As $\rho$ increases, the term $\gamma = 1 - (1 - \rho)(2\mu'\eta - L'\mu'\eta^2)$ approaches 1, slowing down convergence and increasing the steady-state error. Specifically, when $\rho = 0$, the convergence occurs at the fastest rate, with $\gamma = 1 - (1 - \rho)(2\mu'\eta - L'\mu'\eta^2)$, and the convergence speed primarily depends on $\eta$ and $\mu'$. In this case, the steady-state error is smaller, and convergence is faster.

However, as $\rho$ approaches 1, the convergence slows due to the diminished effect of the decay factor $\gamma$. The optimal value of $\rho$ balances the trade-off between convergence speed and steady-state error: smaller values of $\rho$ lead to faster convergence and smaller steady-state error, while larger values lead to slower convergence and higher steady-state error. Furthermore, the steady-state error $\Delta_*$ is influenced by the Lipschitz constant $L'$ of the loss function. A larger $L'$ increases the steady-state error, indicating that the loss function is more sensitive to changes in parameters. To reduce steady-state error, it is desirable to have a smaller $L'$.

To ensure stability, the learning rate $\eta$ must satisfy:

$$0 < \eta < \min\left\{\frac{2}{L'}, \frac{1}{2\mu'(1-\rho)}\right\}, \quad \text{(Lemma } A.3\text{)}$$

which guarantees that $\gamma < 1$ and ensures geometric convergence. If $\eta$ exceeds this threshold, updates may overshoot, leading to instability and preventing convergence.

In conclusion, the recurrence relation is:

$$\Delta_{k+1} = \gamma \Delta_k + \frac{\rho L' \sigma^{(k)^2}}{2},$$

with the steady-state value:

$$\Delta_* = \frac{\rho L' \sigma^{(k)^2}}{2(1 - \rho)(2\mu'\eta - L'\mu'\eta^2)}.$$

and the general solution:

$$\Delta_k = (\Delta_0 - \Delta_*)\gamma^k + \Delta_*.$$

The convergence rate is geometric with rate:

$$\gamma = 1 - (1 - \rho)(2\mu'\eta - L'\mu'\eta^2).$$

The algorithm converges with complexity:

$$\mathcal{O}\left(\gamma^k\right) \Rightarrow \mathcal{O}\left(\left(1 - (1 - \rho)(2\mu'\eta - L'\mu'\eta^2)\right)^k\right).$$

**Lemma A.3.** *Assume that the function $\mathcal{L}(\boldsymbol{\theta}^{(k)})$ is both $L'$-Lipschitz continuous and $\mu'$-strongly convex. Let $\rho$ be the sampling probability with $\gamma = 1 - (1 - \rho)(2\mu'\eta - L'\mu'\eta^2)$ being the decaying rate. From this point onwards, to ensure convergence, the learning rate $\eta$ must satisfy*

$$0 < \eta < \min\left\{ \frac{2}{L'}, \frac{1}{2\mu'(1-\rho)} \right\}.$$

*Proof.* To ensure convergence, the following expression $\gamma = 1 - (1 - \rho)(2\mu'\eta - L'\mu'\eta^2)$ must strictly lie between 0 and 1, we require

$$0 < \gamma < 1.$$

This leads to two inequalities.

1. **Upper Bound ($\gamma < 1$)**

$$\gamma < 1, \text{or}$$
$$1 - (1 - \rho)(2\mu'\eta - L'\mu'\eta^2) < 1, \text{or equivalently}$$
$$-(1 - \rho)(2\mu'\eta - L'\mu'\eta^2) < 0.$$

   Since $(1 - \rho) > 0$ (as $0 \le \rho < 1$),

$$2\mu'\eta - L'\mu'\eta^2 > 0, \text{or}$$
$$\mu'\eta(2 - L'\eta) > 0.$$

   Since $\mu' > 0$ and $\eta > 0$, we require:

$$2 - L'\eta > 0 \quad \Rightarrow \quad \eta < \frac{2}{L'}.$$

   Thus, the upper bound is

$$\eta < \frac{2}{L'}.$$

   —

2. **Lower Bound ($\gamma > 0$)**

$$\gamma > 0, \text{or}$$
$$1 - (1 - \rho)(2\mu'\eta - L'\mu'\eta^2) > 0, \text{or equivalently}$$
$$(1 - \rho)(2\mu'\eta - L'\mu'\eta^2) < 1.$$

   As before, since $(1 - \rho) > 0$,

$$2\mu'\eta - L'\mu'\eta^2 < \frac{1}{1 - \rho}.$$

   Let

$$f(\eta) = 2\mu'\eta - L'\mu'\eta^2.$$

   This is a concave quadratic with maximum at

$$\eta^* = \frac{1}{L'}.$$

   The maximum value is

$$f(\eta^*) = \frac{\mu'}{L'}.$$

Hence, we require

$$\frac{\mu'}{L'} < \frac{1}{1-\rho},$$

which gives a condition on the parameters:

$$\mu' < \frac{L'}{1-\rho}.$$

If this holds, then for all $0 < \eta < \frac{2}{L'}$, the inequality is satisfied.

—

3. **Case when $\frac{\mu'}{L'} \geq \frac{1}{1-\rho}$** If instead $\frac{\mu'}{L'} \geq \frac{1}{1-\rho}$, then we must solve explicitly:

$$2\mu'\eta - L'\mu'\eta^2 < \frac{1}{1-\rho}.$$

Equivalently:

$$L'\mu'\eta^2 - 2\mu'\eta + \frac{1}{1-\rho} > 0.$$

The roots of

$$L'\mu'\eta^2 - 2\mu'\eta + \frac{1}{1-\rho} = 0$$

are

$$\eta = \frac{1 \pm \sqrt{1 - \frac{L'}{\mu'(1-\rho)}}}{L'}.$$

For real roots, we need

$$\mu'(1-\rho) \geq L'.$$

In this case, since the quadratic opens upward, the inequality holds for

$$\eta < \frac{1 - \sqrt{1 - \frac{L'}{\mu'(1-\rho)}}}{L'} \quad \text{or} \quad \eta > \frac{1 + \sqrt{1 - \frac{L'}{\mu'(1-\rho)}}}{L'}.$$

But combined with $\eta < \frac{2}{L'}$, the lower branch is the relevant range.

—

4. **Simplified Practical Bound** In practice, to avoid complexity, we ensure

$$\frac{\mu'}{L'} < \frac{1}{1-\rho},$$

which is usually satisfied if $\rho$ is not too close to 1 and $\mu'$ is not too large relative to $L'$.

Thus, a safe step-size bound is

$$0 < \eta < \min\left\{\frac{2}{L'}, \frac{1}{2\mu'(1-\rho)}\right\}.$$

This ensures both conditions are met:

- $\eta < \frac{2}{L'}$ from $\gamma < 1$,
- $\eta < \frac{1}{2\mu'(1-\rho)}$ from $\gamma > 0$

5. **Final Result:**

$$\boxed{0 < \eta < \min\left\{\frac{2}{L'}, \frac{1}{2\mu'(1-\rho)}\right\}}$$

$\square$

# B   Technical Details

Table 12: Technical specifications across various tasks.

| Task | Models | Datasets | Metrics | Backbone / Component |
|---|---|---|---|---|
| Image Classification | ResNet-50 (He et al., 2016) 
 Swin-T (Liu et al., 2021) 
 MLP-Mixer (Tolstikhin et al., 2021) | CIFAR-10 (Krizhevsky & Hinton, 2009) 
 CIFAR-100 (Krizhevsky & Hinton, 2009) 
 Tiny ImageNet (Le & Yang, 2015) 
 ImageNet (Deng et al., 2009) | *Acc @1* 
 *Acc @5* | — |
| Object Detection | YOLOv7 (Wang et al., 2023) 
 RT-DETRv2 (Lv et al., 2024) | Pascal VOC 2012 (Everingham et al., 2010) | *mAP @0.5* 
 *mAP @0.5:0.95* | — |
| Image Segmentation | U-Net (Ronneberger et al., 2015) 
 Segmenter-Tiny (Strudel et al., 2021) | ADE20K (Zhou et al., 2017) | *SS-IoU* 
 *MS-IoU* | U-Net: ResNet-34 (He et al., 2016) 
 Segmenter: ViT (Dosovitskiy et al., 2021) encoder + MaskFormer (Cheng et al., 2022) decoder |
| Domain Adaptation (DA) | MDD (Zhang et al., 2019) 
 MCC (Jin et al., 2020) | Office-31 (Saenko et al., 2010) | *Acc @1* | ResNet-50 (He et al., 2016) |
| Domain Generalization (DG) | VREx (Krueger et al., 2021) 
 GroupDRO (Sagawa et al., 2020) | PACS (Li et al., 2017) | *Acc @1* | ResNet-50 (He et al., 2016) |
| Federated Learning (FL) | FedAvg (McMahan et al., 2017) 
 FedProx (Li et al., 2020b) 
 FedSpeed (Sun et al., 2023) | CIFAR-10 (Krizhevsky & Hinton, 2009) | *Acc @1* | ResNet-50 (He et al., 2016) |
| LLM Pre-training | GPT-2 (Radford et al., 2019) | IMDB (Maas et al., 2011) | Accuracy 
 Precision 
 Recall 
 F1-Score | — |

# C   Implementation Details

**Image Classification:**

| Image Classification Task | |
|---|---|
| **Repository:** | https://github.com/microsoft/Swin-Transformer |
| **Implementations:** | Integrated with various models and datasets |
| **Total Epochs:** | 200 (Default: 300) |
| **Warmup Epochs:** | 1 (Default: 20) |
| **Trials:** | 3 |
| **AdaDelta Optimizer** | |
| **Learning Rate ($\eta$):** | 0.1 |
| **Warmup Epochs:** | 20 |
| **AdaGrad Optimizer** | |
| **Learning Rate ($\eta$):** | Default Value |
| **Warmup Epochs:** | 20 |

Table 13: Hyperparameters for the Image Classification Task

**Object Detection**

| Object Detection Task | |
|---|---|
| **Repositories:** | **YOLOv7:** https://github.com/WongKinYiu/yolov7 |
| | **RT-DETRv2:** https://github.com/lyuwenyu/RT-DETR |
| **Epochs:** | YOLOv7: 100, RT-DETRv2: 72 (Default) |

Table 14: Hyperparameters for the Object Detection Task

**Image Segmentation**

| Image Segmentation Task | |
|---|---|
| **Repository:** | https://github.com/rstrudel/segmenter |
| **Models:** | Segmenter (Strudel et al., 2021), U-Net (Ronneberger et al., 2015) |
| **Training Epochs:** | 20 (Default) |

Table 15: Hyperparameters for the Image Segmentation Task

**Domain Adaptation (DA) and Domain Generalization (DG)**

| Domain Adaptation (DA) and Domain Generalization (DG) | |
|---|---|
| **Repository:** | https://github.com/thuml/Transfer-Learning-Library |
| **Methods:** | MDD (Zhang et al., 2019), MCC (Jin et al., 2020), Vrex (Krueger et al., 2021), GroupDRO (Sagawa et al., 2020) |
| **Training Schedule:** | 200 iterations per epoch, 20 epochs |

Table 16: Hyperparameters for Domain Adaptation and Domain Generalization Experiments

**Federated Learning:**

| Federated Learning (FL) | |
|---|---|
| **Repository:** | https://github.com/woodenchild95/FL-Simulator |
| **Methods:** | FedAvg (McMahan et al., 2017), FedProx (Li et al., 2020b), FedSpeed (Sun et al., 2023) |

Table 17: Hyperparameters for Federated Learning (FL) Experiments

For all our experiments, we kept other parameters unchanged.

## D   Actual vs Effective Epochs

Table 18 compares the number of **epochs** under different **sampling strategies**, highlighting the **efficiency gains** achieved through sampling. For example, with **Pe = 5** over **200 epochs**, the training involves **167 backpropagations** and **33 sampled iterations**, whereas the baseline **without sampling** performs **167 backpropagations** and **0 sampled steps**. Performance is reported across **CIFAR-10**, **CIFAR-100**, and **Tiny ImageNet** datasets as **(with sampling, without sampling)**.

## E   Experiments with Different Activation Functions

As shown in Table 19, our proposed sampling strategy demonstrates consistent effectiveness across various activation functions. Despite reducing the number of training epochs, the strategy maintains competitive accuracy levels, indicating its robustness and generalizability in different architectural settings.

Table 18: Comparison of **backpropagation operations** with and without **sampling strategies**, demonstrating **efficiency gains** across CIFAR-10, CIFAR-100, and Tiny ImageNet over 200 epochs. Reported metrics follow the format **(with sampling, without sampling)**, and the **higher value in each tuple is bolded**.

| Model | # Effective epochs | CIFAR-10 | | CIFAR-100 | | TINY | |
|---|---|---|---|---|---|---|---|
| | | *Acc @1(↑)* | *Acc @5(↑)* | *Acc @1(↑)* | *Acc @5(↑)* | *Acc @1(↑)* | *Acc @5(↑)* |
| ResNet-50 | Pe = 5 (167) | (**90.69**,88.53) | (**99.67**,97.21) | (**68.62**,66.54) | (**90.53**,88.83) | (**65.40**,63.24) | (**85.60**,83.73) |
| | Pe = 10 (183) | (**91.29**,89.02) | (**99.78**,97.82) | (**68.90**,66.93) | (**90.98**,89.31) | (**66.00**,63.85) | (**86.00**,84.23) |
| | Pr = 0.2 (176) | (**90.87**,88.82) | (**99.66**,97.53) | (**69.36**,67.25) | (**90.87**,89.05) | (**65.86**,63.92) | (**86.24**,84.56) |
| | Pr = 0.5 (135) | (**90.16**,88.27) | (**99.68**,97.65) | (**66.86**,64.58) | (**90.06**,88.45) | (**63.48**,61.23) | (**84.84**,83.07) |
| | Pr = 0.7 (119) | (**89.32**,87.04) | (**99.57**,97.02) | (**64.97**,63.26) | (**89.25**,87.53) | (**62.10**,60.08) | (**84.08**,82.34) |
| Swin-T | Pe = 5 (167) | (**85.17**,83.52) | (**99.30**,96.54) | (**64.66**,62.73) | (**88.77**,86.81) | (**63.08**,61.04) | (**84.96**,83.14) |
| | Pe = 10 (183) | (**86.36**,84.85) | (**99.41**,96.91) | (**65.36**,63.65) | (**89.11**,87.34) | (**64.52**,62.57) | (**85.22**,83.54) |
| | Pr = 0.2 (176) | (**85.99**,84.53) | (**99.35**,96.83) | (**65.72**,63.82) | (**88.97**,87.02) | (**63.60**,61.85) | (**85.04**,83.35) |
| | Pr = 0.5 (135) | (**83.71**,82.04) | (**99.14**,96.23) | (**62.06**,60.25) | (**87.53**,85.54) | (**61.50**,59.58) | (**83.60**,81.83) |
| | Pr = 0.7 (119) | (**81.67**,80.07) | (**98.99**,96.04) | (**59.67**,57.84) | (**86.11**,84.02) | (**59.44**,57.23) | (**82.22**,80.15) |
| MLP-Mixer | Pe = 5 (167) | (**81.47**,79.53) | (**99.00**,95.82) | (**55.88**,54.03) | (**82.75**,81.04) | (**53.60**,52.04) | (**77.48**,76.07) |
| | Pe = 10 (183) | (**81.79**,79.83) | (**99.02**,95.91) | (**57.27**,55.57) | (**83.84**,81.85) | (**55.00**,53.35) | (**78.76**,77.13) |
| | Pr = 0.2 (176) | (**81.18**,79.34) | (**98.98**,95.75) | (**57.24**,55.38) | (**83.58**,81.64) | (**54.02**,52.53) | (**78.30**,76.52) |
| | Pr = 0.5 (135) | (**80.27**,78.58) | (**98.83**,95.54) | (**54.06**,52.27) | (**81.71**,80.02) | (**51.70**,50.04) | (**76.38**,75.03) |
| | Pr = 0.7 (119) | (**78.67**,77.04) | (**98.70**,95.24) | (**51.59**,50.03) | (**80.08**,78.34) | (**49.24**,47.83) | (**75.06**,73.54) |

Table 19: Acc@1 and Acc@5 (%) with different sampling strategies across activation functions (*ReLU, Sigmoid, Tanh*). Values are bolded if they exceed the corresponding baseline values.

| Strategy | Acc@1 (%) | | | Acc@5 (%) | | |
|---|---|---|---|---|---|---|
| | *ReLU* | *Sigmoid* | *Tanh* | *ReLU* | *Sigmoid* | *Tanh* |
| Baseline | 98.06 | 95.68 | 97.90 | 99.95 | 99.82 | 99.96 |
| Pe = 5 | 98.03 | 95.17 | **97.97** | **99.96** | **99.88** | **99.97** |
| Pe = 10 | **98.07** | **95.91** | **97.96** | **99.96** | **99.89** | **99.98** |
| Pr = 0.2 | **98.15** | **95.92** | **97.98** | **99.96** | **99.87** | **99.98** |
| Pr = 0.5 | 97.89 | 94.87 | 97.62 | **99.96** | **99.85** | **99.98** |
| Pr = 0.7 | 97.86 | 94.37 | 97.48 | 99.95 | 99.74 | 99.90 |

## F   Random Noise vs GGD

we experimented by replacing GGD with some random noise below $\mathcal{N}(0, 0.01)$ and reported in the Table 20. The table below clearly shows that our method maintains superior performance, even when replaced with random noise.

## G   Analysing the Error Margins

We analyze the error margins by applying our sampling strategies on **Tiny-ImageNet** across 15 independent trials, as reported in Table 21. The results reveal that the observed error margins remain consistently tight,

Table 20: CIFAR-100 accuracies (Top-1, Top-5) and Random Noise performance for different strategies on ResNet-50 and Swin-T. Baseline is highlighted, best Top-1 and Top-5 per model are bolded. Random Noise reports lower performance than standard evaluation.

| Models | Strategy (#effective iterations) | GGD (Top-1, Top-5) | $\mathcal{N}(0,0.01)$ (Top-1, Top-5) |
|---|---|---|---|
| ResNet-50 | **Baseline (71,168)** | (69.96, 91.64) | - |
| | Pe = 5 (65,297) | (68.02, 90.39) | (65.10, 88.50) |
| | Pe = 10 (71,553) | (**69.98**, **91.86**) | (67.20, 90.30) |
| | Pr = 0.2 (68,816) | (68.42, 91.77) | (66.50, 90.00) |
| | Pr = 0.5 (52,785) | (65.90, 89.71) | (63.80, 87.90) |
| | Pr = 0.7 (46,529) | (64.21, 89.05) | (61.50, 86.20) |
| Swin-T | **Baseline (71,168)** | (64.53, 88.58) | - |
| | Pe = 5 (65,297) | (61.95, 87.05) | (59.10, 84.50) |
| | Pe = 10 (71,553) | (**64.86**, **88.66**) | (62.80, 86.90) |
| | Pr = 0.2 (68,816) | (64.58, 88.59) | (62.00, 86.30) |
| | Pr = 0.5 (52,785) | (57.99, 84.96) | (55.50, 82.10) |
| | Pr = 0.7 (46,529) | (55.41, 83.25) | (53.00, 80.50) |

demonstrating the stability of our approach. More importantly, even when several epochs are skipped due to the sampling strategies, the performance does not degrade compared to the baseline for both **ResNet-50** and **Swin Transformer**. This consistency across multiple trials highlights that our method not only reduces the number of effective backpropagation operations but also achieves this efficiency **without compromising accuracy** or reliability.

Table 21: Swin Transformer on Tiny-ImageNet over 15 trials, reporting Top-1 and Top-5 accuracies. Narrow confidence intervals (0.10–0.25) support reliable conclusions.

| Strategy | Accuracy (Top-1, Top-5) |
|---|---|
| Baseline | $(63.12 \pm 0.18, 85.01 \pm 0.22)$ |
| Pe = 5 | $(62.58 \pm 0.21, 83.65 \pm 0.17)$ |
| Pe = 10 | $(\mathbf{63.42 \pm 0.14, 85.21 \pm 0.20})$ |
| Pr = 0.2 | $(\mathbf{63.31 \pm 0.23, 85.19 \pm 0.19})$ |
| Pr = 0.5 | $(59.34 \pm 0.20, 81.95 \pm 0.24)$ |
| Pr = 0.7 | $(59.91 \pm 0.16, 81.26 \pm 0.21)$ |
| DP=5 | $(61.89 \pm 0.22, 84.05 \pm 0.18)$ |
| DP=10 | $(\mathbf{63.15 \pm 0.13, 85.25 \pm 0.16})$ |
| DR=0.2 | $(\mathbf{63.15 \pm 0.20, 85.23 \pm 0.15})$ |
| DR=0.5 | $(61.42 \pm 0.17, 84.06 \pm 0.22)$ |
| DR=0.7 | $(60.83 \pm 0.19, 83.61 \pm 0.24)$ |

## H   Analyzing the Training Loss Plots

From the loss plots in Figure 4, obtained using the Swin-Transformer on CIFAR-10 with sampling performed every 10 epochs, the proposed training loss (orange) consistently remains slightly below or closely matches the original loss, reflecting a modest improvement in convergence speed. Likewise, the proposed validation loss (orange) is generally lower than the original (blue) across most epochs, indicating that the proposed approach not only accelerates convergence but also enhances generalization.

## I   Analyzing the GGD

From Figure 5, the successive differences of weight updates in the Swin-Transformer model on the CIFAR-10 dataset—sampled at intervals of 10 epochs—exhibit a unimodal distribution when evaluated over subsets of

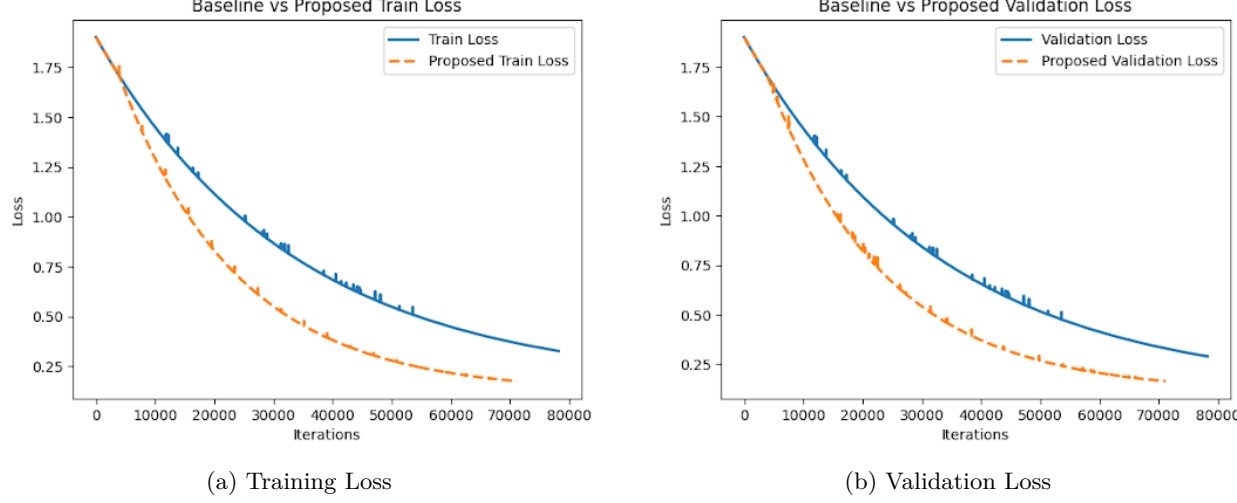

(a) Training Loss

(b) Validation Loss

Figure 4: Training and validation loss curves on CIFAR-100 comparing the original baseline with the proposed sampling method, evaluated at intervals of 10 epochs.

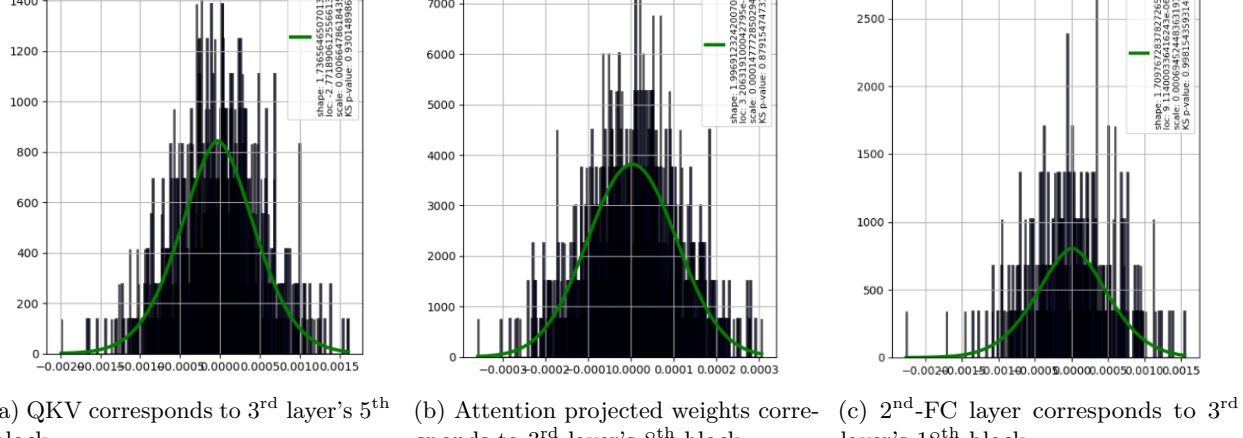

(a) QKV corresponds to $3^{rd}$ layer's $5^{th}$ block.

(b) Attention projected weights corresponds to $3^{rd}$ layer's $8^{th}$ block.

(c) $2^{nd}$-FC layer corresponds to $3^{rd}$ layer's $18^{th}$ block.

Figure 5: Distribution of successive epoch weight differences across layer subsets, compared with the GGD envelope (green).

elements from each layer. These differences closely align with the Generalized Gaussian Distribution (GGD) envelope (shown in green), indicating that the GGD provides an effective characterization of the weight update dynamics across layers.

