# OpenReview forum: "Energy-Efficient Deep Learning via Update Sampling from a Generalized Gaussian Distribution: An Empirical Study"
_TMLR — Rejected by TMLR_

### Review · Reviewer_uKoW · 2025-07-18

**Summary Of Contributions:**

This paper proposes a strategy of noise injection when training deep neural networks. First, the authors propose to model the noise over the update of each parameter with a Generalized Gaussian Distribution (GGD) denoted by $p$. Then, they propose to replace some of the updates proposed by the training algorithm (SGD, Adam, etc.) with updates randomly sampled w.r.t. the distribution $p$. Since sampling updates w.r.t. $p$ is less costly than performing a backprop, this strategy is expected to be more energy-efficient than the original training algorithm (i.e., computationally cheaper).

Experiments are meant to show that this strategy leads to better accuracies than the original training algorithms, although it requires less backprop passes.

The authors also provide a convergence speed for their strategy when the loss is strongly convex.

**Audience:**

Yes

**Claims And Evidence:**

No

**Requested Changes:**

**Clarity:**
* state the proposed method as a noise injection method, remove statements claiming that the method estimates the update, the gradient, etc.
* homogenize notation, remove notation that appear only once, simplify cumbersome notation;
* improve the readability of the figures.

**Mathematics:**
* fix the error in the proof of Theorem 3.1;
* if possible, state the theorem with more realistic assumptions (the loss is far from being strongly convex when training neural networks).

**Experiments:**
* propose at least one series of experiments where the error margins allow for a significant conclusion (possibly by performing much more experiments in one setup);
* **comparing only with the baseline is very limited**: compare with other noise-injection methods;
* check and fix the values in the tables.

**Strengths And Weaknesses:**

# Strengths
## Clarity
Overall, the paper is easy to understand.

## Relevance
Reducing the computational cost of neural network training is of high interest for the community.

## Significance
Although the experimental results are not groundbreaking, the proposed idea is worth exploring.

# Weaknesses
## Clarity
**Idea of the method.** According to the abstract, "energy savings are achieved by skipping entire training epochs and estimating gradients by sampling from a GGD". This is not true. In practice, the sampled vector is only noise, whose mean, scale and tail parameters are estimated layer-wise. In particular, the mean is not intended to be close to the true gradient. This way of presenting the method is confusing: the gradients are not estimated, they are replaced by noise. So, the entire method consists in incorporating noise during the training process.

**Inconsistent use of bold letters.** At the first glance, it seems that bold letters are used for vectors ($\boldsymbol{\theta}, \boldsymbol{\delta}$). But some inconsistencies occur:
* Eqn. (6): $\mu$ is subtracted to $\boldsymbol{\delta}$ : what does it mean? Is $\mu$ a scalar that is subtracted to each coordinate of $\boldsymbol{\delta}$ (which is an abuse of notation)? Or is $\mu$ a vector (the bold is not used consistently).
* Eqn. (9): it is certain that $\xi$ is a vector, so it should be in bold.

**Undefined/inconsistent notation:**
* In Section 3.2, Eqn. (4): stricly speaking, this equation does not make sense: a vector $\boldsymbol{\theta}$ is sampled from a distribution that apparently outputs a scalar. But there is no clear way to be sure: is it a multivariate GGD? if it is a univariate GGD, should we infer that the coordinates of $\boldsymbol{\theta}$ are sampled independently (which is a strong assumption)? Understanding this part is harder when we expect that the GGD is meant to "_estimate_ gradients" (and not injecting noise).
* Eqn. (6): what is the norm $| \cdot |$?
* Notations $\boldsymbol{\delta}^{(k)}_l$ and $\boldsymbol{\delta}^{(k)}|_l$ seem to be used indistinctly, as well as $\boldsymbol{\theta}^{(k)}_l$ and $\boldsymbol{\theta}^{(k)}|_l$. I would recommend to choose one of these once and for all (and I would prefer $\boldsymbol{\theta}^{(k)}_l$ because it is less cumbersome).
* I would not recommend to denote by $\sigma$ the scale parameter of a GGD, since it can be confounded with the standard deviation. This mistake is done in the paramgraph 'Image classification" (p. 8).
* The letter $B$ (for minibatch) is sometimes used as index, sometimes as exponent.

In general, the nature of the objects is sometimes ambiguous. Consistent notation rules should be adopted, along with a clear mention of the nature of the objects, i.e., write somewhere that $\boldsymbol{\xi} \in \mathbb{R}^n$ and $\mu \in \mathbb{R}$, etc.

**Figures.** Figures 2 and 3, and Table 2 are difficult to read. In particular, on these figures, the legend, the scale and the titles are impossible to read (especially when printed), and the contrast between the empirical distribution (in black) and the theoretical curve (green/red) is too low.

The captions related to Figures 2, 3 contain several typos: "at at" or "at at at"…

Hyperparameters $h_1, \cdots, h_p$ and training algorithm $\mathcal{G}$: these notations are used only once, so they should be easily removable. Using only $\delta$ should be sufficient.

## Technical aspects
**Theorem 3.1.** First, it is surprising to read $\mathbb{E}[\hat{\boldsymbol{\delta}}^{(k)}] \approx 0$ in the hypotheses of the theorem. If the statement is informal, it should be indicated somewhere.

Second, the proof lacks rigor. Essentially, one can read, in the sequence of equations (17):
* we have: $$a \leq b + (1 - 2 \mu \eta + L \mu \eta^2) \Delta,$$where $\Delta \geq 0$;
* since $\eta \ll 0$, then $\eta^2 \approx 0$ (I assume that the authors meant $\eta \ll 1$);
* then: $$a \leq b + (1 - 2 \mu \eta) \Delta .$$
Which is mathematically false.

Additionally, the two last lines of this series of equations are identical.

**Experimental results.** The experimental results are put in different tables. Each accuracy has been averaged on 3 runs, and some error margin has been added (I assume that it is the standard deviation over the 3 runs). Results are in bold if they are the best of their series of experiments. However, there is no consideration for the error margins. In fact, in all the tables, most of the results in bold overlap with the baseline if we take into account the error margins. So, the results are mainly inconclusive.

Additionally, in Table 2, column 1 (CIFAR-10, Acc @1), row MLP-Mixer/DR=0.2, the accuracy of 81.22 is in bold, although it is lower than the baseline, 81.99.

---

> ### Author Response · Authors · 2025-09-05
> **Response for uKoW**
>
> We thank the reviewer for the careful evaluation of our work and the valuable feedback. Specifically, we are grateful for bringing out the strengths (``high interest for the community``, ``the proposed idea is worth exploring``) and weaknesses (``way presenting the method is confusing``, ``some inconsistencies``, ``the proof lacks rigor``) of our work. Addressing these comments has helped us improve the technical quality and the presentation quality of the manuscript. Further, the additional experiments suggested by the reviewer have helped provide more justification. We hope that our responses adequately address the concerns.
>
> **Idea of the method:** The parameters of the GGD are obtained from the differences between consecutive parameter updates. Using maximum likelihood estimation (MLE) on these differences, we estimate the mean, scale, and shape parameters, and then sample from this fitted distribution. These samples replace full backpropagation while retaining the statistical characteristics of the observed update differences. Our method does not skip entire epochs; rather, it reduces the overall **FLOPs** required for training by substituting backpropagation with statistically informed sampling steps.
> **Significance of $\mu$:** In practice, $\mu$ acts as a scalar offset applied to each element of $\boldsymbol{\delta}^{(k)}_\ell$ prior to sampling.
>
> **Clarity in equation (4)** We acknowledge that our notations were not clear and we regret for that. We have updated the revised manuscript with clear notations. Coming to equation (4) we model $p(\boldsymbol{\theta^{(k)}}_l | p(\boldsymbol{\theta^{(k-1)}}_l) ) = GGD(\mu^{(k)}_l, \sigma^{(k)}_l, \beta^{(k)}_l)$.
> For tractability, we assume a univariate GGD, where each coordinate is sampled independently from the fitted distribution. Parameters are estimated via MLE on $\boldsymbol{\delta}_l^{(k)}$, repeated $n_l$ times, and reshaped to match $\boldsymbol{\delta}_l^{(k)}$.  While this independence assumption is strong, it substantially simplifies computation and, as shown in experiments, does not harm performance. The shape parameter $\beta_l^{(k)}$ further enables modeling both heavy- and light-tailed behaviors.
>
> **B as minibatch instance:** We use $B$ consistently to denote the mini-batch index. Specifically, $\boldsymbol{\xi}_\ell^B$ refers to stochastic noise, defined as the deviation between the mini-batch gradient and the deterministic gradient. We acknowledge notational inconsistencies and have  fixed them in the revised manuscript.
>
> **Comparing only with the baseline is very limited:** Beyond the baseline, we include comparisons with **MeZO** (NeurIPS 2023), a recent noise-injection method that estimates gradients using symmetric perturbations via two forward passes (with batch size $B$ and perturbation scale $\epsilon$). Results are reported in Table~11 of the revised manuscript, situating our method within the context of state-of-the-art noise-based optimization techniques. Also we experimented by replacing GGD with some random noise below 𝒩(0,0.01) and reported in the Table 20 in the revised manuscript. The table below clearly shows that our method maintains superior performance, even when replaced with random noise.
> |Models|Strategy|GGD(Top-1,Top-5)|𝒩(0,0.01)(Top-1,Top-5)|
> |---|---|---|---|
> |ResNet-50|Baseline|(69.96,91.64)|-|
> ||Pe=5|(68.02,90.39)|(65.10,88.50)|
> ||Pe=10|(69.98,91.86)|(67.20,90.30)|
> ||Pr=0.2|(68.42,91.77)|(66.50,90.00)|
> ||Pr=0.5|(65.90,89.71)|(63.80,87.90)|
> ||Pr=0.7|(64.21,89.05)|(61.50,86.20)|
> |Swin-T|Baseline|(64.53,88.58)|-|
> ||Pe=5|(61.95,87.05)|(59.10,84.50)|
> ||Pe=10|(64.86,88.66)|(62.80,86.90)|
> ||Pr=0.2|(64.58,88.59)|(62.00,86.30)|
> ||Pr=0.5|(57.99,84.96)|(55.50,82.10)|
> ||Pr=0.7|(55.41,83.25)|(53.00,80.50)|
>
>
> **Improve the readability of the figures:** We have improved figure readability by enlarging font sizes, clarifying axis labels, and explicitly reporting location, scale, and shape parameters in captions. Each figure can now be interpreted independently.
>
> **State the proposed method as a noise injection method:** We have revised the manuscript to consistently describe our approach as a _noise injection method_. All prior claims of “gradient estimation” have been removed or rephrased. The emphasis is now placed on injecting statistically modeled noise derived from historical update differences. We thank the reviewer for the careful observation.
>
> **Experiments where the error margins allow for a significant conclusion:** We conducted experiments using the Swin Transformer on the Tiny-ImageNet dataset over 15 independent trials. The results, summarized in the Table 21 (revised manuscript), demonstrate narrow error margins that enable statistically significant conclusions: These results show consistent performance across configurations, with strategies such as $Pe=10$, $Pr=0.2$, $DP=10$, and $DR=0.2$ yielding the strongest outcomes. The narrow confidence intervals validate the robustness of our findings.

---

> > ### Author Response · Authors · 2025-09-05
> > **Experiments with error margins**
> >
> > **Experiments where the error margins allow for a significant conclusion:** We conducted experiments using the Swin Transformer on the Tiny-ImageNet dataset over 15 independent trials. The results, summarized in the Table 21 (revised manuscript), demonstrate narrow error margins that enable statistically significant conclusions: These results show consistent performance across configurations, with strategies such as $Pe=10$, $Pr=0.2$, $DP=10$, and $DR=0.2$ yielding the strongest outcomes. The narrow confidence intervals validate the robustness of our findings.
> >
> > | **Strategy** | **Accuracy (Top-1, Top-5)** |
> > |--------------|------------------------------|
> > | Baseline     | (63.12 ± 0.18, 85.01 ± 0.22) |
> > | Pe=5         | (62.58 ± 0.21, 83.65 ± 0.17) |
> > | Pe=10        | **(63.42 ± 0.14, 85.21 ± 0.20)** |
> > | Pr=0.2       | **(63.31 ± 0.23, 85.19 ± 0.19)** |
> > | Pr=0.5       | (59.34 ± 0.20, 81.95 ± 0.24) |
> > | Pr=0.7       | (59.91 ± 0.16, 81.26 ± 0.21) |
> > | DP=5         | (61.89 ± 0.22, 84.05 ± 0.18) |
> > | DP=10        | **(63.01 ± 0.13, 85.25 ± 0.16)** |
> > | DR=0.2       | **(63.05 ± 0.20, 85.23 ± 0.15)** |
> > | DR=0.5       | (61.42 ± 0.17, 84.06 ± 0.22) |
> > | DR=0.7       | (60.83 ± 0.19, 83.61 ± 0.24) |

---

> > > ### Comment · Reviewer_uKoW · 2025-09-08
> > > **Remarks about the math**
> > >
> > > I acknowledge the authors' answer, and I would like to provide more details about the math (Theorem 3.1).
> > >
> > > **Hypotheses.** One reads: $\mathbb{E}[\hat{\boldsymbol{\delta}}^{(k)}] \approx 0$
> > > in the hypotheses. This assumption is not acceptable in a mathematical proof. The assumption must be precise: either $\mathbb{E}[\hat{\boldsymbol{\delta}}^{(k)}] = 0$ or $|\mathbb{E}[\hat{\boldsymbol{\delta}}^{(k)}]| \leq c$ with hypotheses on $c$.
> > > The reader must not be left free to interpret this hypothesis.
> > >
> > > **Proof (between equations 15 and 16).** The exact same remark holds about $L' \mu' \eta^2 \approx 0$, which I already highlighted in my review. To be precise, the authors **must** provide precise bounds on $L'$ and $\eta$ to continue the proof. $\eta$ and $L'$ should be small enough, but compared to what? Moreover, the authors assume that $L' \ll 1$, which is both imprecise and not mentioned in the hypotheses of the theorem. The issue at stake is: what is the range of $\eta$ for which the result of Theorem 3.1 holds?
> > >
> > > This issue is not unimportant, since "TMLR emphasizes technical correctness over subjective significance".

---

> ### Author Response · Authors · 2025-09-05
> **Theoretical justification**
>
> $\mathcal{L}(\boldsymbol{\theta}^{(k+1)}) \leq \mathcal{L}(\boldsymbol{\theta}^{(*)}) + (1-2\mu'\eta+L'\mu'\eta^2)$, where $L'$ and $\mu'$ denote the Lipschitz and strong convexity parameters (notation revised in the manuscript for consistency). Since $\eta$ and $L'$ are both much smaller than $1$, the term $L'\mu'\eta^2$ can be regarded as negligible, i.e., $L'\mu'\eta^2 \approx 0$.

---

> ### Author Response · Authors · 2025-09-10
> **Correction to Theorem, Lemmas A.1 and A.2**
>
> We sincerely regret our oversight of your careful and important comments. We are grateful to you for checking our work again and reminding us of TMLR's emphasis on technical correctness. Based on your feedback, we have made the following corrections to the assumptions in Lemma A.1 and the proof of Lemma A.2. These changes will be reflected in the revised manuscript.
>
> ## Correction to Theorem
> Let $\rho$ be the sampling probability, and suppose that the update error $\boldsymbol{\theta}^{(k)}$ follows a Generalized Gaussian Distribution (GGD), with $\mathbb{E}[\boldsymbol{\theta}^{(k)}] = \boldsymbol{0}$ and $\mathbb{E}[\|\boldsymbol{\theta}^{(k)}\|^2] \leq \sigma^{(k)^2}$.
> Assume the loss function $\mathcal{L}(\boldsymbol{\theta})$ is $\mu'$-strongly convex and $L'$-Lipschitz continuous, with optimal parameter $\boldsymbol{\theta}^{(*)}$. If the learning rate satisfies $0 < \eta < \min\left\\{ \frac{2}{L'},  \frac{1}{2\mu'(1-\rho)} \right\\}$, then the expected loss difference:
>
> $\Delta_k= \mathbb{E}[\mathcal{L}(\boldsymbol{\theta}^{(k)}) - \mathcal{L}(\boldsymbol{\theta}^{(*)})]$ satisfies: $\Delta_{k+1} \leq \gamma \Delta_k + \frac{\rho L' \sigma^{(k)^2}}{2},$
>
> where $ \gamma = 1 - (1-\rho)(2\mu'\eta - L'\mu'\eta^2) $. Consequently, the convergence rate is $\mathcal{O}(\gamma^k)$.
>
> ## Correction to Lemma A.1
> **Lemma A.1:** Assume that the function $\mathcal{L}({\theta^{(k)}})$ is $L'$-Lipschitz continuous, with $\mathbb{E} [{\hat{\boldsymbol{\delta}}^{(k)}}] = \boldsymbol{\mu}^{(k)} = \boldsymbol{0}$.
>
> The proof of the above lemma remains the same.
>
> ## Correction to Lemma A.2
> **Lemma A.2** Assume that the function $\mathcal{L}(\boldsymbol{\theta}^{(k)})$ is both $L'$-Lipschitz continuous and $\mu'$-strongly convex. Then $\boldsymbol{\theta}^{(k+1)}$ follows the SGD update rule:
> $\mathbb{E}[{\mathcal{L}({\boldsymbol{\theta}^{(k+1)}})}] \leq \mathbb{E}[{\mathcal{L}({\boldsymbol{\theta}^{(*)}})}] + (1 - 2\mu' \eta + L'\mu'\eta^2) \Delta_k, $ where
>
> $\Delta_k=\mathbb{E}\left[\mathcal{L}(\boldsymbol{\theta}^{(k)}) - \mathcal{L}(\boldsymbol{\theta}^{(*)})\right],$
>
> and $\boldsymbol{\theta}^{(*)}$ is the optimal parameter. In the corrected proof, the term $L'\mu'\eta^2$ is no longer ignored. Further, $L'$ is no longer assumed to be much smaller than 1. However, the learning rate $\eta$ is still assumed to be orders of magnitude smaller than 1 . As flagged by the reviewer in the initial review, this change also fixes the erroneous inequality between (16) and (17). This is accounted for in the corrected expression for $\gamma$. Specifically,
>
> $$
> \gamma = 1 - (1-\rho)(2\mu'\eta - L'\mu'\eta^2) = 1 -\mu'\eta(1-\rho) (2 - L'\eta).
> $$

---

> ### Author Response · Authors · 2025-09-10
> **Bound on $\eta$**
>
> To ensure convergence, the following expression
> $
> \gamma = 1 - (1-\rho)(2\mu'\eta - L'\mu'\eta^2)
> $
> must strictly lie between $0$ and $1$, we require
> $$
> 0 < \gamma < 1.
> $$
> This leads to two inequalities.
>
> ## Upper Bound ($\gamma < 1$)
> $$
> \gamma  < 1, \text{or}
> $$
>
> $$
> 1 - (1-\rho)(2\mu'\eta - L'\mu'\eta^2) < 1, \text{or equivalently}
> $$
>
> $$
> -(1-\rho)(2\mu'\eta - L'\mu'\eta^2) < 0.
> $$
>
> Since $(1-\rho) > 0$ (as $0 \leq \rho < 1$),
>
> $$
> 2\mu'\eta - L'\mu'\eta^2 > 0, \text{or}
> $$
>
> $$
> \mu'\eta (2 - L'\eta) > 0.
> $$
>
> Since $\mu' > 0$ and $\eta > 0$, we require:
> $$
> 2 - L'\eta > 0
> \quad \Rightarrow \quad
> \eta < \frac{2}{L'}.
> $$
>
> Thus, the upper bound is
> $$
> \eta < \frac{2}{L'}.
> $$
>
> ---
>
> ## Lower Bound ($\gamma > 0$)
>
> $$
> \gamma  > 0, \text{or}
> $$
>
> $$
> 1 - (1-\rho)(2\mu'\eta - L'\mu'\eta^2) > 0, \text{or equivalently}
> $$
>
> $$
> (1-\rho)(2\mu'\eta - L'\mu'\eta^2) < 1.
> $$
>
> As before, since $(1-\rho) > 0$,
> $$
> 2\mu'\eta - L'\mu'\eta^2 < \frac{1}{1-\rho}.
> $$
>
> Let
> $$
> f(\eta) = 2\mu'\eta - L'\mu'\eta^2.
> $$
> This is a concave quadratic with maximum at
> $$
> \eta^\ast = \frac{1}{L'}.
> $$
>
> The maximum value is
> $$
> f(\eta^\ast) = \frac{\mu'}{L'}.
> $$
>
> Hence, we require
> $$
> \frac{\mu'}{L'} < \frac{1}{1-\rho},
> $$
> which gives a condition on the parameters:
> $$
> \mu' < \frac{L'}{1-\rho}.
> $$
>
> If this holds, then for all $0 < \eta < \tfrac{2}{L'}$, the inequality is satisfied.
>
> ---
>
> ## Case when $\tfrac{\mu'}{L'} \geq \tfrac{1}{1-\rho}$
> If instead $\tfrac{\mu'}{L'} \geq \tfrac{1}{1-\rho}$, then we must solve explicitly:
> $$
> 2\mu'\eta - L'\mu'\eta^2 < \frac{1}{1-\rho}.
> $$
>
> Equivalently:
> $$
> L'\mu'\eta^2 - 2\mu'\eta + \frac{1}{1-\rho} > 0.
> $$
>
> The roots of
> $$
> L'\mu'\eta^2 - 2\mu'\eta + \frac{1}{1-\rho} = 0
> $$
> are
> $$
> \eta = \frac{1 \pm \sqrt{1 - \tfrac{L'}{\mu'(1-\rho)}}}{L'}.
> $$
>
> For real roots, we need
> $$
> \mu'(1-\rho) \geq L'.
> $$
>
> In this case, since the quadratic opens upward, the inequality holds for
> $$
> \eta < \frac{1 - \sqrt{1 - \tfrac{L'}{\mu'(1-\rho)}}}{L'}
> \quad \text{or} \quad
> \eta > \frac{1 + \sqrt{1 - \tfrac{L'}{\mu'(1-\rho)}}}{L'}.
> $$
>
> But combined with $\eta < \tfrac{2}{L'}$, the lower branch is the relevant range.
>
> ---
>
> ## Simplified Practical Bound
> In practice, to avoid complexity, we ensure
> $$
> \frac{\mu'}{L'} < \frac{1}{1-\rho},
> $$
> which is usually satisfied if $\rho$ is not too close to $1$ and $\mu'$ is not too large relative to $L'$.
>
> Thus, a safe step-size bound is
> $$
> 0 < \eta < \min\left\\{ \frac{2}{L'},  \frac{1}{2\mu'(1-\rho)} \right\\}.
> $$
>
> This ensures both conditions are met:
>
> * $\eta < \tfrac{2}{L'}$ from $\gamma < 1$,
> * $\eta < \tfrac{1}{2\mu'(1-\rho)}$ from $\gamma > 0$
>
> This in turn results in the following expression for the steady state $\Delta_*$. Recall that
> $\Delta_* = \frac{\rho L'\sigma^{(k)^2}}{2(1 - \gamma)}. \Rightarrow \Delta_* = \frac{\rho L' \sigma^{(k)^2}}{2(1 - \rho)(2\mu'\eta - L'\mu'\eta^2)} . $
>
> With these changes, and as implemented in all experiments across different tasks, the learning rate $\eta$ is scheduled according to a predefined criterion (see below Table), where $\eta_0$ denotes the initial learning rate and $\eta_k$ the learning rate at epoch $k$). Consequently, $\eta$ gradually decays and eventually becomes much smaller than 1 ($\eta \ll 1$). Therefore, our earlier observations on convergence, steady-state $\Delta_*$, and transient $\Delta_k$ continue to hold. Moreover, our empirical estimation of the Lipschitz constant $L'$ was found to be in the range $(0.5, 0.99)$.
>
> | **Task**        | **Model**          | **Initial Learning Rate** ($\eta_0$) | **Learning Rate Schedule** ($\eta = \eta_k$) |
> |-----------------|--------------------|---------------------------------------|----------------------------------------------|
> | Classification  | Swin Transformer   | $5 \times 10^{-4}$                    | $\eta_k = \eta_0 \cdot \left( \gamma' + \tfrac{1-\gamma'}{2} \left[ 1 + \cos\left( \tfrac{\pi (k \bmod K_{\text{cycle}})}{K_{\text{cycle}}} \right) \right] \right)$ ; $K_{\text{cycle}} = 30$ |
> | Detection       | YOLOv7             | $0.01$                                | $\eta_k = \eta_0 \cdot \left( \gamma' + \tfrac{1-\gamma'}{2} \left[ 1 + \cos\left( \tfrac{\pi (k \bmod K_{\text{cycle}})}{K_{\text{cycle}}} \right) \right] \right)$ ; $K_{\text{cycle}} = 3$ |
> | Detection       | RT-DETRv2          | $2 \times 10^{-4}$                    | $\eta_k = \eta_0 \cdot \gamma^{\text{step}}$ ; $\gamma^{\text{step}} = 0.1$ |
> | Segmentation    | Segmenter          | $10^{-3}$ (ADE20K, Pascal-Context), $10^{-2}$ (Cityscapes) |$\eta_k = \eta_0 \left(1 - \dfrac{\mathrm{iter}_k}{\mathrm{totalIter}}\right)^{0.9}$
>  |
> | FL              | ResNet-18          | $0.01$                                | $0.01$ |
>
> Here, $\gamma'$ and $\gamma^{\text{step}}$ are decay parameters, and $K_{\text{cycle}}$ denotes the cycle period (in epochs) over which the schedule is applied.

---

### Review · Reviewer_ev5h · 2025-08-12

**Summary Of Contributions:**

The authors propose an energy-efficient approach to training deep learning models by estimating gradients via sampling from the Generalized Gaussian Distribution (GGD) and periodically skipping the backpropagation step. This results in a lossy but more efficient training process. They evaluate their method in diverse settings, including federated learning, domain generalization/adaptation, and across various tasks such as image classification, object detection, and LLM pre-training.

**Audience:**

Yes

**Broader Impact Concerns:**

I do not have any concerns on the ethical implications of the work.

**Claims And Evidence:**

Yes

**Requested Changes:**

See weakness / questions written above.

**Strengths And Weaknesses:**

Strengths:
- The writing is clear and the explanations are easy to follow.
- The authors provide a detailed justification of their hypothesis (that the gradient follows a GGD) using the Kolmogorov–Smirnov test.
- The method is evaluated on a comprehensive range of settings and benchmark tasks.

Weakness / Questions:
- It would be helpful to see results in terms of real GPU-hour savings, not just total FLOPs saved.
- I am curious about the cost of fitting the GGD at each iteration (does the fitting data size increase in later iterations?). However, I do not expect this to be significantly more expensive.

Overall, I think this paper presents a fairly novel idea accompanied by broad and in-depth analysis. I hope the authors address the questions I have raised.

---

> ### Author Response · Authors · 2025-09-05
> **GPU computational time**
>
> We thank the reviewer for the careful evaluation of our work and the valuable feedback. Specifically, we are grateful for bringing out the strengths (``writing is clear``, `` detailed justification of their hypothesis``, ``method is evaluated on a comprehensive range of settings and benchmark tasks``) and weaknesses/questions (``results in terms of real GPU-hour savings, not just total FLOPs saved.``, ``cost of fitting the GGD at each iteration ``) of our work. Addressing these comments by conducting the suggested additional experiments has helped us improve the technical quality and provide more relevant data points as justification. We hope that our responses adequately address the concerns.
>
> **Real GPU-hour saving**
> To quantify actual computational benefits, we measured GPU time via the number of \textit{effective backpropagations} (i.e., gradient computations and updates). In our method, full backpropagation is performed only in early epochs, while later epochs are replaced with sampling steps. This reduces the total number of effective backprops, which directly lowers GPU time. As shown in the below table, reducing effective backprops from 200 to 135 in ResNet-50 lowers training time from $8$H:$25$M to $4$H:$23$M (a $\sim$48\% saving). Similarly, Swin-T shows savings of up to $\sim$50\%. Thus, substituting late-epoch backpropagation with sampling achieves substantial GPU-hour reductions while retaining comparable accuracy.
>
> | **Models** | **Strategy (#effective backprops)** | **(Top-1, Top-5)** | **Time** |
> |------------|-------------------------------------|---------------------|----------|
> | **ResNet-50** | **Baseline (200)** | (69.96, 91.64) | 8H:25M |
> |            | Pe = 5 (167)  | (68.02, 90.39) | 7H |
> |            | Pe = 10 (183) | **(69.98, 91.86)** | 7H:53M |
> |            | Pr = 0.2 (176) | **(69.42, 91.77)** | 7H:5M |
> |            | Pr = 0.5 (135) | (65.90, 89.71) | 4H:23M |
> |            | Pr = 0.7 (119) | (64.21, 89.05) | 2H:38M |
> | **Swin-T** | **Baseline (200)** | (64.53, 88.58) | 10H:45M |
> |            | Pe = 5 (167)  | (61.95, 87.05) | 7H:5M |
> |            | Pe = 10 (183) | **(64.86, 88.66)** | 7H:53M |
> |            | Pr = 0.2 (176) | **(64.58, 88.59)** | 7H:10M |
> |            | Pr = 0.5 (135) | (57.99, 84.96) | 4H:23M |
> |            | Pr = 0.7 (119) | (55.41, 83.25) | 2H:40M |
>
> **Table:** CIFAR-100 accuracies (Top-1, Top-5) and computation time for different strategies on ResNet-50 and Swin-T. Baseline is highlighted, and best Top-1 and Top-5 per model are bolded.

---

### Review · Reviewer_piNj · 2025-08-19

**Summary Of Contributions:**

This work introduces the `GradSample` technique to reduce energy consumption when training neural networks.
Specifically, this algorithm regularly skips *exact backpropagation* steps and instead:
1. fits a generalized Gaussian distribution (GGD) using previous gradient updates, and
2. draws a sample from this model to use as a cheaper surrogate update.

Assuming that gradient updates follow a GGD, which the authors show
empirically, they prove an upper bound on the expected loss with this approach
and a corresponding convergence rate.  The authors empirically compare
`GradSample` for different sampling schedules to standard SGD on a variety of
tasks including image classification, object detection, image segmentation,
federated learning, and LLM pretraining.

**Audience:**

Yes

**Claims And Evidence:**

Yes

**Requested Changes:**

**Typos and suggestions**
- [page 03] Suggestion: Define the GGD distribution before using it in Eq. (4) in Section 3.2.
  One clean approach is to define GGD by itself in Section 3.2 and then start the "Hypothesis"
  subsection as Section 3.3.
- [page 03] Typo: "In other words from 3 at each ..." --> "In other words from (3) at each ..."
- [page 03] Suggestion: It would improve readavility to boldface $\mu^{(k)}$ since it is a vector.
  Further, the norm in Eq. (6) for $|\mathbf{\delta}_{\ell}^{(k)} - \mathbf{\mu}^{(k)}|$ should be defined.
- [page 04] Suggestion: In Theorem 3.1, consider boldfacing $\mathbf{0}$ since this is the zero vector.
- [page 05] Typo: "the convergence trend," --> "the convergence trend."
- [page 05] Suggestion: In Table 1, consider replacing "Equation" with "Update Rule"
- [page 08] Typo: "layers of CNNs ," --> "layers of CNNs,"
- [page 12] Typo: "datasets.Notably," --> "datasets. Notably,"
- [page 19] Suggestion: The font size in Table 13 doesn't match the rest of the paper.
- [page 21] Suggestion: The font size in Table 19 doesn't match the rest of the paper.

**Strengths And Weaknesses:**

**Strengths**
- Clean idea: At every $k$-th epoch, the `GradSample` algorithm estimates the
  gradient update via a learned GGD, draw sample(s), and uses them
  instead of the more expensive (true) gradient for that epoch.
- Expeirments show that `GradSample` outperforms baseline if sample gradients
  are used [0.1, 0.2] percent of the time. This is likely a nice form of regularization
  while also being cheaper, assuming GGD is cheap to estimate and draw samples.

**Weaknesses**
- Assumes that all $L$ layers in the neural network follow the same
  distribution. Is this reasonable?  For example, in a transformer block should
  the self-attention layer and subsequent MLP layers obey this property?
- The energy savings of this method is questionable. For example, if we use
  sampled gradients in every other step and we assume estimating the parameters
  of the GGD for that step is free, we only reduce the energy consumption by 50%.

**Questions**
- [page 04] Do all losses considered satisfy the $\mathbf{L}$-Lipschitz
  continuity? If so, what are the values of $\mathbf{L}$?
- [page 04] How expensive is it each time you estimate the parameters of the
  GGD? How does this compare to the back propagation you're aiming to replace?
- [page 05] How exactly is $\gamma$ defined in Theorem 3.1? $\mu$ is a vector
  but it's being treated as a scalar.
- [page 05] What are the actual values in Figure 1? Are they based on a
  particular model/dataset? If so, say which experiment. Or are they universal
  values and this just highlights a gap in your analyis? If it's the latter,
  why isn't the analysis tight?
- [page 06] In Figure 2 and Figure 3, even though these distributions are
  unimodal and appear to be a GGD, why is it reasonable to assume that each
  entry of $\mathbf{\delta}^{(k)}$ is i.i.d.? It seems reasonable that entries
  should be correlated.
- [page 08] Do the FLOP counts in Table 2 account for estimating the params of
  the GGD each time the sampling condition is met?
- [page 11] How do adaptive optimizers that use values of $\theta^{(k-1)}$ work
  if this gradient is sampled? Do you just treat it as unsampled?

---

> ### Author Response · Authors · 2025-09-05
> **Addressing notations and GPU computational time (with sampling, without sampling)**
>
> We thank the reviewer for the careful evaluation of our work and the valuable feedback. Specifically, we are grateful for bringing out the strengths (``Clean idea``, ``likely a nice form of regularization``) and weaknesses (``Assumes that all $L$ layers in the neural network follow the same distribution``, ``The energy savings of this method is questionable.``) of our work. The list of questions and typos is particularly appreciated. Addressing these comments by conducting additional experiments and fixing the typos has helped us improve the technical quality and the presentation quality of the manuscript. We hope that our responses adequately address the concerns.
>
> **Significance of $\gamma$ and $\boldsymbol{\mu}$:** The notation $\boldsymbol{\mu}$ in the original draft was intended to denote $\boldsymbol{\mu}$-strong convexity, but we acknowledge that this caused confusion with the mean $\mu$. To improve clarity, in the updated draft we now use $\mu'$ for the strong convexity parameter and $L'$ for the Lipschitz constant. The parameter $\gamma$ is defined in the theorem as $\gamma = (1 - 2\boldsymbol{\mu}'\eta(1-\rho))$. As $\rho \to 1$, the convergence becomes slower due to the reduced influence of the decay factor $\gamma$. The optimal choice of $\rho$ balances convergence speed and steady-state error: smaller values of $\rho$ yield faster convergence with smaller steady-state error, whereas larger values result in slower convergence and higher steady-state error (further details about the role of each parameter values please check the appendix A in the updated draft).
>
> **Actual values in Figure 1:** Figure1 is obtained from a controlled simulation of the theoretical recursion. In our experiments, we estimate the Lipschitz constant of the loss ($L'$) as the largest eigenvalue (spectral norm) of the Hessian, which typically lies between $0.5$ and $1.0$ on CIFAR-10/100 with ResNet. Accordingly, we fixed $L'=1.0$, $\mu'=1.0$, and varied $\sigma^2 \in {0.01,0.05,0.1,0.5,0.9}$, $\eta \in {0.01,0.05,0.1,0.2,0.3}$, and $\rho \in {0.2,0.5,0.7}$ over 200 iterations, ensuring $\eta < \tfrac{1}{2\mu'(1-\rho)}$. The results show a gap between the theoretical upper bound and actual trajectories, indicating that these values are illustrative rather than \textbf{universal constants}.
>
> **Figure 2 and Figure 3 unimodality GGD:** Even though the histograms in Figures 2 and 3 look unimodal and resemble a GGD, we model each entry as i.i.d. mainly for tractability. In practice, correlations between entries may exist, but accounting for them would require a much more complex model. The i.i.d. assumption is a standard simplification that captures the dominant marginal behavior while keeping the analysis mathematically manageable.
>
> **How expensive is it each time you estimate the parameters of the GGD? How does this compare to the back propagation you're aiming to replace?**
> Estimating the parameters of the GGD is computationally negligible when sufficient resources are available. We conducted experiments and summarized the results in below Table (with sampling, without sampling)  and in revised draft (Table 18), where we compare our sampling strategy with standard backpropagation. For instance, with 200 epochs, the Pe=5 strategy requires 167 backpropagation operations and 33 periodic sampling operations. To ensure fairness, we also experimented with 167 epochs without sampling and compared it with Pe=5. All experiments were conducted on an **_NVIDIA Tesla V100-SXM2-32GB_** GPU, a standard research-grade accelerator widely used in deep learning and high-performance computing.
>
> |**Model**|**# Backprops**|**Acc @1 (↑)**|**Acc @5 (↑)**|**Time (With, Without)**|
> |---|---|---|---|---|
> |**ResNet-50**|Pe = 5 (167)|(**68.62**, 66.54)|(**90.53**, 88.83)|(7H:01M:01S, 7H:00M:59S)|
> ||Pe = 10 (183)|(**68.90**, 66.93)|(**90.98**, 89.31)|(7H:53M:59S, 7H:53M:42S)|
> ||Pr = 0.2 (176)|(**69.36**, 67.25)|(**90.87**, 89.05)|(7H:05M:31S, 7H:05M:07S)|
> ||Pr = 0.5 (135)|(**66.86**, 64.58)|(**90.06**, 88.45)|(4H:24M:29S, 4H:23M:12S)|
> ||Pr = 0.7 (119)|(**64.97**, 63.26)|(**89.25**, 87.53)|(2H:39M:39S, 2H:38M:18S)|
> |**Swin-T**|Pe = 5 (167)|(**64.66**, 62.73)|(**88.77**, 86.81)|(7H:06M:18S, 7H:05M:45S)|
> ||Pe = 10 (183)|(**65.36**, 63.65)|(**89.11**, 87.34)|(7H:54M:07S, 7H:53M:50S)|
> ||Pr = 0.2 (176)|(**65.72**, 63.82)|(**88.97**, 87.02)|(7H:11M:07S, 7H:10M:43S)|
> ||Pr = 0.5 (135)|(**62.06**, 60.25)|(**87.53**, 85.54)|(4H:24M:20S, 4H:23M:15S)|
> ||Pr = 0.7 (119)|(**59.67**, 57.84)|(**86.11**, 84.02)|(2H:41M:33S, 2H:40M:12S)|
>
> From the table, it is evident that our sampling strategy consistently outperforms the baseline, even when compared against actual backpropagation operations. This highlights the effectiveness of our method in maintaining accuracy without any compromise.

---

> ### Author Response · Authors · 2025-09-05
> **Adaptive optimizers**
>
> **How do adaptive optimizers that use values of $\boldsymbol{\theta}^{(k-1)}$ work if this gradient is sampled? Do you just treat it as unsampled?**
> Yes, as described in the algorithm, the model parameters are updated only when the sampling condition is satisfied, by incorporating the sampled updates into the previous parameters. If the condition is not met, we use standard SGD or other adaptive optimizers.
>
> Consider the ADAM update equations:
>
> $$
> \mathbf{g}^{(k)} = \nabla_{\boldsymbol{\theta}^{(k-1)}} \mathcal{L}\big(\boldsymbol{\theta}^{(k-1)}\big)
> $$
>
> $$
> \mathbf{m}^{(k)} = \rho_1 \mathbf{m}^{(k-1)} + (1-\rho_1)\mathbf{g}^{(k)}
> $$
>
> $$
> \mathbf{v}^{(k)} = \rho_2 \mathbf{v}^{(k-1)} + (1-\rho_2)(\mathbf{g}^{(k)})^2
> $$
>
> $$
> \hat{\mathbf{m}}^{(k)} = \frac{\mathbf{m}^{(k)}}{1-\rho_1^{k}}
> $$
>
> $$
> \hat{\mathbf{v}}^{(k)} = \frac{\mathbf{v}^{(k)}}{1-\rho_2^{k}}
> $$
>
> $$
> \boldsymbol{\theta}^{(k)} = \boldsymbol{\theta}^{(k-1)} - \alpha \frac{\hat{\mathbf{m}}^{(k)}}{\sqrt{\hat{\mathbf{v}}^{(k)} + \varepsilon}}
> $$
>
>
> From these update equations, we can express the parameter change as:
>
> $$
> \boldsymbol{\theta}^{(k)} - \boldsymbol{\theta}^{(k-1)}
> = - \left( \alpha \frac{\hat{\mathbf{m}}^{(k)}}{\sqrt{\hat{\mathbf{v}}^{(k)} + \varepsilon}} \right)
>  \sim \text{GGD}\big(\mu^{(k)}, \sigma^{(k)}, \beta^{(k)}\big)
> $$

---

### Author Response · Authors · 2025-09-05
**Thanks to the reviewers for detailed feedback**

We thank all the reviewers for their careful evaluation of our work. We are especially grateful for their appreciation of the idea, their identification of weaknesses, and their constructive suggestions to improve both the technical content and the presentation of our work. We have carefully addressed all comments, incorporated the necessary changes into the revised manuscript, and hope that our responses meet the reviewers’ expectations.

**Summary of Changes in the Revised Manuscript:**
- Corrected grammatical issues.
- Improved clarity and consistency of notations.
- Added additional experiments:
  - Comparison with random Gaussian noise 𝒩(0,0.01).
  - Experiments with Swin-Transformer on Tiny-ImageNet over 15 trials to analyze error margins.
  - Estimation of GPU time by comparing actual vs. sampled backpropagation operations.
- Enhanced readability of figures by adding scale, location, and shape values in each sub-caption, and by providing overall figure-level information (regarding the experiments and architecture) in the main captions (e.g., Figure 2).

---

### Author Response · Authors · 2025-09-15
**Revised manuscript with corrected proof**

## Summary of Changes in the final Revised Manuscript

- **Correction to Lemma A.2:** We have fixed the erroneous inequality between (16) and (17). The corrected expression is now given by $\gamma = 1 - (1 - \rho)(2\mu'\eta - L'\mu'\eta^2)$.

- **Correction to Theorem 3.1:** We have updated the expression from $\mathbb{E}[\boldsymbol{\theta}^{(k)}] \approx \boldsymbol{0}$ to $\mathbb{E}[\boldsymbol{\theta}^{(k)}] = \boldsymbol{0}$. In addition, the conditional bound on the learning rate $\eta$ has been updated to $0 < \eta < \min\left\\{ \frac{2}{L'},  \frac{1}{2\mu'(1-\rho)} \right\\}$, with the proof formally established in **Lemma A.3**.

We sincerely thank the reviewers for their valuable feedback, which has helped us improve the manuscript. These corrections and additions have been incorporated into the uploaded **latest revised manuscript**, and we kindly invite the reviewers to please look at the updated version and share their comments..

---

### Comment · Action_Editor_RuCZ · 2025-09-26
**Questions about the paper**

Dear authors,

I have a few questions about the method that I think would clarify the contribution.

# GGD
In eq. 5, you write that the vector $\delta$ follows the GGD distribution, yet I only see scalars in that distribution. As I understand from your response, you assume that all the scalar parameters in $\delta$ are independent. I think the paper should be much more transparent about this crucial point. This seems like a very strong hypothesis, which is not verified in practice. It is crucial to acknowledge this in the paper. As such, I completely agree with the remark of rev.uKoW: the sampled "gradient approximations" have absolutely no structure (while "true" gradients have a structure...); the proposed method is about injecting noise in the steps rather than trying to approximate gradients. What are your thoughts on the question? I believe the paper should be more transparent about these points.

# Experiments

The proposed method can be seen as a new optimization algorithm. As such, we expect to see faster convergence. Yet, there are no training curves. I expect the proposed method to minimize the loss faster, as this is the premise upon which the paper is based.

I would also expect another baseline, which simply repeats gradient steps (i.e., instead of skipping an iteration by sampling from GGD, simply re-use the previously computed gradient).

# Notation

Below Eq.3, you write $\delta = \mathcal{G}(\nabla L(\theta^{k-1}), h_1, \dots h_p)$, and state that this can model adam or SGD. This is wrong: Adam needs a memory of former gradients, and SGD needs stochastic gradients. Likewise, the current framing cannot take decoupled weight decay into account. Please update accordingly.

This also leads to a misunderstanding regarding Simsekli et al. 2019. In that paper, the authors mention the structure of the gradients, not that of the Adam updates, which can be drastically different.

# Epoch
What you can epoch in the paper is usually called "iteration". An epoch is usually understood as a pass over the whole dataset. Please update accordingly.

Thanks for your insights,
AE

---

> ### Author Response · Authors · 2025-10-01
> **Response to AE Questions**
>
> We sincerely thank the AE for these important questions and the new baseline suggestion. Answering these questions has helped us improve the quality of the presentation and provide crucial technical details. All these changes are reflected in the revised manuscript.
>
> **Another baseline, which simply repeats gradient steps.**
> We experimented with the suggested baseline using the Swin Transformer with different sampling strategies on the CIFAR-100 dataset. Specifically, each time our approach samples from the GGD, we repeat the previous parameter values (instead of adding the GGD samples). The results of three different trials are tabulated below.
>
> | **Strategy** | **Top-1 Acc (%)** | **Top-5 Acc (%)** | **Suggested Top-1 (%)** | **Suggested Top-5 (%)** |
> |--------------|-------------------|-------------------|--------------------------|--------------------------|
> | Pe= 5   | 61.95 ± 1.83 | 87.05 ± 1.44 | 59.87 ± 1.80 | 83.12 ± 1.42 |
> | Pe= 10  | 64.86 ± 1.54 | 88.66 ± 1.05 | 61.92 ± 1.52 | 85.33 ± 1.03 |
> | Pr= 0.2 | 64.58 ± 1.01 | 88.59 ± 1.62 | 62.48 ± 1.00 | 86.01 ± 1.60 |
> | Pr= 0.5 | 57.99 ± 1.76 | 84.96 ± 1.64 | 55.12 ± 1.74 | 82.45 ± 1.62 |
> | Pr= 0.7 | 55.41 ± 1.03 | 83.25 ± 4.05 | 53.80 ± 1.02 | 80.90 ± 4.00 |
> | DPe= 5  | 61.13 ± 0.37 | 86.72 ± 0.23 | 58.40 ± 0.36 | 83.21 ± 0.23 |
> | DPe= 10 | 64.57 ± 0.19 | 88.96 ± 0.16 | 61.89 ± 0.19 | 85.90 ± 0.16 |
> | DPr= 0.2 | 64.99 ± 0.06 | 88.78 ± 0.36 | 62.35 ± 0.06 | 87.01 ± 0.35 |
> | DPr= 0.5 | 60.39 ± 0.22 | 85.95 ± 0.28 | 57.88 ± 0.22 | 82.77 ± 0.28 |
> | DPr= 0.7 | 59.60 ± 0.04 | 85.82 ± 0.25 | 56.85 ± 0.04 | 83.50 ± 0.25 |
>
> Our sampling strategy outperforms this baseline in terms of accuracy. In the interest of time, this experiment was conducted on one model (swin transformer) with a relatively smaller dataset (CIFAR-100).
>
> **Equation 5:** We apologize for the sloppiness in our notation. We have now modified (5) as follows. At each layer $\ell$ with $n_\ell$ elements,  $\boldsymbol{\theta}^{(k)}_{\ell, i}$ denotes the $i^{\text{th}}$ element of the parameter vector $\boldsymbol{\theta}^{(k)}\_\ell$
>
>
> $P(\boldsymbol{\theta}^{(k)}\_{\ell, i} | \boldsymbol{\theta}^{(k-1)}\_{\ell, i})  \sim \text{GGD}(\mu^{(k)}\_{\ell}, \sigma^{(k)}\_{\ell}, \beta^{(k)}\_{\ell})$
>
>
> In other words
>
> $\boldsymbol{\delta}^{(k)}_{\ell,i} = (\boldsymbol{\theta}^{(k)}\_{\ell,i}-\boldsymbol{\theta}^{(k-1)}\_{\ell,i})\sim\text{GGD}(\mu^{(k)}\_\ell,\sigma^{(k)}\_\ell, \beta^{(k)}\_\ell)$
>
> As noted in the response **uKoW**, we sampled $n_\ell$ elements from the GGD fitted for $\boldsymbol{\delta}^{(k)}_\ell$.
>
>
>
> Although this independence assumption is indeed a strong one, it substantially simplifies computation and, as shown in experiments, does not harm performance. The shape parameter $\beta_\ell^{(k)}$ further enables modeling both heavy- and light-tailed behaviors.
>
>
> Furthermore, our empirical investigation of the distribution of the individual update elements $\boldsymbol{\delta}^{(k)}_{\ell,i}$ revealed a unimodal histogram centered at zero, which can be accurately modeled with a GGD. These histograms are included in Appendix I of the revised manuscript.
>
> **Equation 3:** We agree that our notation is erroneous for the identified cases. We have removed the term $\boldsymbol{\delta}^{(k)} = \mathcal{G}(\nabla \mathcal{L}(\boldsymbol{\theta}^{(k-1)}, h_1, h_2, \ldots, h_p))$, and simply let $\boldsymbol{\delta}^{(k)}$ denote the parameter update at iteration $k$.
>
>
> **Training plots:** Thanks for this nice suggestion. Our model converges faster, requiring fewer iterations, where each iteration corresponds to a single mini-batch update. These training plots are included in Appendix H of the revised manuscript.
>
>
> **Iteration vs. Epoch:** We appreciate this important clarification. As noted, a mini-batch update corresponds to an iteration, while an epoch refers to a complete pass over the dataset, consisting of $B$ mini-batches. In our method, updates are indexed by the epoch count $k$, so the total number of mini-batch updates up to $\boldsymbol{\theta}^{(k)}$ is $k \times B$. Thus, skipping an *epoch* in this context effectively corresponds to skipping an *iteration* at the end of $k$ epochs. This distinction does not alter our reported results or the measured energy savings.

---

### Decision · Action_Editor_RuCZ · 2025-10-15

**Recommendation:** Reject

**Audience:**

Yes

**Audience Explanation:**

Having ways to accelerate neural network training is, for sure, interesting to many TMLR's readers.

**Claims And Evidence:**

No

**Claims Explanation:**

This paper presents an algorithm to skip gradient updates in the training loop of neural networks.
It does so by generating a random gradient, drawn from a distribution that assumes that all the entries of the gradients are i.i.d., and fits 1-d distributions to match these distributions. The authors argue that the distribution of real gradient updates can be modeled as i.i.d. noise, and that sampling from this estimated distribution can approximate the effects of gradient descent while saving computation. The approach is evaluated across a wide range of domains and shows comparable final accuracies with reduced FLOPs compared to standard training.


# Strengths
- The topic is timely, and methods that accelerate training are always interesting.
- The method is very generic and can be applied to virtually any ML training loop.
- The experimental validation covers many topics.
- The idea of using statistical properties of gradients to skip updates is interesting and could inspire future works.

# Weaknesses
- The current framing of the paper makes the reader believe that this random noise is an approximation to actual gradients, but it cannot have any structure, while gradients are structured.
- The empirical analysis focuses mainly on final test accuracy. To substantiate the claimed energy savings, the paper should include training loss vs. compute or time curves that demonstrate comparable optimization progress under matched FLOPs budgets. It would make it much more straightforward to see how the proposed algorithm helps accelerate learning.
In other words, while the central claim of the paper is that Algorithm 1 achieves a similar train loss as standard training for a wide range of sampling conditions, at a reduced cost, it is never clearly demonstrated.
- For instance, in tables 2 and 3, we don't know what happens if we reduce the baseline's training time to match the flops of the proposed method.
- The mathematical justification is weak: the “GGD hypothesis” is mostly descriptive, and its connection to the proposed update rule is heuristic.
- Some aspects of the presentation are confusing, e.g., Algorithm 1 suggests that updates occur once per epoch rather than per iteration.

Overall, the paper proposes an interesting idea that could have some potential. Still, it requires a substantial reframing to 1) present the proposed method as a stochastic update method rather than a method that tries to imitate gradient steps and 2) give a more convincing demonstration of the benefits of the proposed method in terms of training speed.

**Resubmission Of Major Revision:**

The authors may consider submitting a major revision at a later time.